evolution

ageing, intergenerational effects, Lansing effect, parental age effects, senescence

**Author for correspondence:**
Laura M. Travers
e-mail: laura.travers@uea.ac.uk

†These authors contributed equally to this study.

# Beneficial cumulative effects of old parental age on offspring fitness

Laura M. Travers[1,†], Hanne Carlsson[1,†], Martin I. Lind[2] and Alexei A. Maklakov[1]

[1]School of Biological Sciences, University of East Anglia, Norwich Research Park, Norwich NR4 7TJ, UK
[2]Department of Ecology and Genetics, Animal Ecology, Uppsala University, Norbyvägen 18D, 75236 Uppsala, Sweden

 LMT, 0000-0002-0004-8261; AAM, 0000-0002-5809-1203

Old parental age is commonly associated with negative effects on offspring life-history traits. Such parental senescence effects are predicted to have a cumulative detrimental effect over successive generations. However, old parents may benefit from producing higher quality offspring when these compete for seasonal resources. Thus, old parents may choose to increase investment in their offspring, thereby producing fewer but larger and more competitive progeny. We show that *Caenorhabditis elegans* hermaphrodites increase parental investment with advancing age, resulting in fitter offspring who reach their reproductive peak earlier. Remarkably, these effects increased over six successive generations of breeding from old parents and were subsequently reversed following a single generation of breeding from a young parent. Our findings support the hypothesis that offspring of old parents receive more resources and convert them into increasingly faster life histories. These results contradict the theory that old parents transfer a cumulative detrimental 'ageing factor' to their offspring.

## 1. Introduction

The influence of parental age on offspring phenotypic quality has been widely reported in many taxa [1–3] and is the subject of renewed theoretical [4–7] and empirical interest [8–10]. Adverse effects of parental age have been known for a long time. A study of seventeenth century early American settlers found that children of older mothers lived shorter lives than those born to younger mothers [11]. Later, Albert Lansing's work on parthenogenic rotifers [12,13] (see also [14]) showed that selection lines propagated through older parents gave rise to shorter-lived offspring and went extinct faster than selection lines using young parents. Subsequent studies have confirmed that offspring of older mothers may have reduced lifespan (a phenomenon often called the Lansing effect) in a wide range of taxa, including flies [8,15], nematodes [3], butterflies [16], water fleas [17], birds [18–21], mice [22] squirrels [23] and further work on pre-industrial humans [24].

The lifespan reduction in offspring of older parents led Lansing to suggest that age-dependent changes, or an 'ageing factor', is passed on to offspring of older parents, shortening the lives of those offspring. In addition to offspring lifespan, several studies across various taxa show an age-associated decline in several other offspring traits such as embryo viability [25], development rate [26], larval [27] and juvenile [28] survival. Several mechanisms have been proposed to explain adverse age-related effects on offspring quality, such as a decline in gamete quality, somatic deterioration in parents leading to physiological decline of the reproductive system, and reduction in parental care or investment that may negatively affect the offspring (for review, see [6]).

Although the negative consequences of advanced parental age on offspring are widely documented, in some contexts older parents produce higher quality offspring because of having more resources to invest in offspring and/or more experience or engage in increased reproductive effort in later life [29–32].

Several studies have shown that increased parental age does not necessarily lead to lower quality offspring [33]. A recent study on wild yellow-bellied marmots (*Marmota flaviventer*) found that daughters born to older mothers had higher annual reproductive success and consequently higher lifetime reproductive success than daughters born to younger mothers [34]. In great tits (*Parus major*), offspring of older mothers senesce faster, but also reproduce more in early life leading to overall lifetime fitness similar to offspring of young mothers [18]. Similarly, Plaistow *et al.* [17] found that water flea offspring from older parents had higher early-life reproduction, reached their reproductive peak earlier and senesced faster. Thus, while offspring of older parents may be shorter-lived, increased resource provisioning may result in offspring with altered life-history trajectories, characterized by increased early-life reproduction, that compensate for the longevity cost or are even beneficial for offspring fitness.

Nevertheless, the circumstances under which parental age positively or negatively influences offspring life-history trajectories and fitness remain unclear. Whether the effects are positive or negative also has important implications for evolutionary theories of senescence. For instance, age-related declines in offspring fitness should reduce the relative value of late-life reproduction and lead to the evolution of faster ageing rates, and thus shorter lifespan, while positive age-related parental effects are predicted to increase the strength of selection in later life, leading to slower rates of ageing and longer lifespan [5,7]. While classical theories on the evolution of ageing have assumed that the fitness of offspring is independent of parental age, there has been renewed theoretical interest in how offspring quality varies as a function of parental age [4,5,7,24].

To explore the long-term evolutionary consequences of parental age effects, we investigated multigenerational effects of parental age from young or old parents on offspring fitness in the self-fertilizing hermaphroditic nematode *Caenorhabditis elegans*. To do this, we calculated individual fitness in offspring from lines of worms propagated over multiple generations from *young* and *old* parents, respectively. Unmated *C. elegans* hermaphrodites live for approximately three weeks under standard laboratory conditions; however, they only reproduce for the first 5 days of adulthood, using their own sperm supply (self-fertilization) [35]. Their reproductive period can be extended by mating with males [36]. However, males (and thus mating events) are very rare owing to infrequent meiotic non-disjunction which produces males at a frequency of 0.1–0.2% [37,38]. Given that natural populations consist almost exclusively of hermaphrodites that are incapable of inseminating each other, self-fertilization is the main mode of reproduction [35]. During the 5 days of reproduction, most offspring are produced in the first 2 days of adulthood, followed by a steep decline on day 3, before ceasing by the end of day 5 [39]. In this study, we selected 1-day old self-fertilizing (i.e. non-mated) hermaphrodites as the *young* parental age and 3-day old adults to represent *old* parental age and assessed offspring fitness in the first, third and sixth successive generation. This approach allowed us to test for cumulative inter-generational effects of parental age. After six generations, we tested whether cumulative parental age effects were fully reversible. Finally, we further explored how parental age alters specific life-history traits in offspring by measuring egg size, development time, body size at sexual maturity, lifetime reproduction and lifespan of offspring.

# 2. Material and methods

## (a) Nematode worms

We used Bristol N2 wild-type *C. elegans* nematodes in all assays. In nature, *C. elegans* are mostly found as self-fertilizing hermaphrodites [37,38]. Before the start of the experiment, we bleached plates to collect eggs from worms recovered from frozen stocks. To remove any trans-generational parental age effects, we maintained worms for three generations after thawing, by conducting egg layings with 2-day old adults in each generation.

We used standard nematode growth medium (NGM) agar plates to grow the nematode populations [37], with added antibiotics (100 µg ml$^{-1}$ ampicillin and 100 µg ml$^{-1}$ streptomycin) and a fungicide (10 µg ml$^{-1}$ nystatin) to avoid infections [40]. Before the experiment began, we fed the nematode populations antibiotic resistant *Escherichia coli* OP50-1 (pUC4 K), gifted by J. Ewbank at the Centre d'Immunologie de Marseille-Luminy, France. From defrosting and throughout the experiment, we kept worms in climate chambers at 20°C and 60% relative humidity. During the assays, we kept all worms on 35 mm plates and with 0.2 ml of the *E. coli* seeding suspension. After the start of each experiment, we placed individual eggs on fresh NGM plates to remove possible effects of density-dependence as well as sibling and offspring-parent interactions [41].

## (b) Cumulative parental effects over six generations assay

### (i) Parental age propagation regimes: young, old and switched
To investigate the cumulative effect of parental age on offspring lifespan and reproduction, we set up three parental age regimes. We took offspring from unmated (i.e. self-fertilizing) adult hermaphrodites at the first day of reproduction (*young* parental propagation regime) and on the third day of reproduction when reproduction declines steeply (*old* parental propagation regime) for six consecutive generations. We selected 1-day old adults as the *young* parental age because it coincides with the start of reproduction. Three-day old adults were selected to represent *old* parental age because most offspring are produced in the first 2 days. Day 3 is also the last day when offspring production is sufficiently high to provide offspring to maintain several generations of propagation (see age-specific reproduction data in Results). We also included a *switched* propagation regime to assess the reversibility of cumulative transgenerational effects owing to old parental age. In the *switched* parental propagation regime, which also ran for six generations, we took offspring from three generations of old parents (3-day old adults) followed by three generations of offspring from young parents (1-day old adults). See figure 1 for schematic of cumulative parental effects assay.

To create the three parental propagation regimes (*young*, *old* and *switched*), three offspring were taken from 33 2-day old hermaphrodites (a total of 99 offspring) to set up 33 lines of the three propagation regimes. We used a paired design so that offspring from each of the original 33 parents were represented in all three parental propagation regimes (see figure 1 for schematic of experimental design). During six generations, we picked eggs from 1-day old parents and from 3-day old parents in the *young* and *old* parental age propagation regimes, respectively. In the *switched* regime, we picked eggs from 3-day old parents in the first three generations and then from 1-day old parents from generation four to six.

On the day of setting up a new generation (1- or 3-day old adult), we conducted a three-hour egg laying to produce age synchronized eggs. After 3 h, we placed two haphazardly selected eggs onto individual plates. We set up two eggs to have one backup in case one of the eggs did not hatch. Only one of the offspring was used to propagate the line. We conducted egg layings in each generation at the same time of day within each

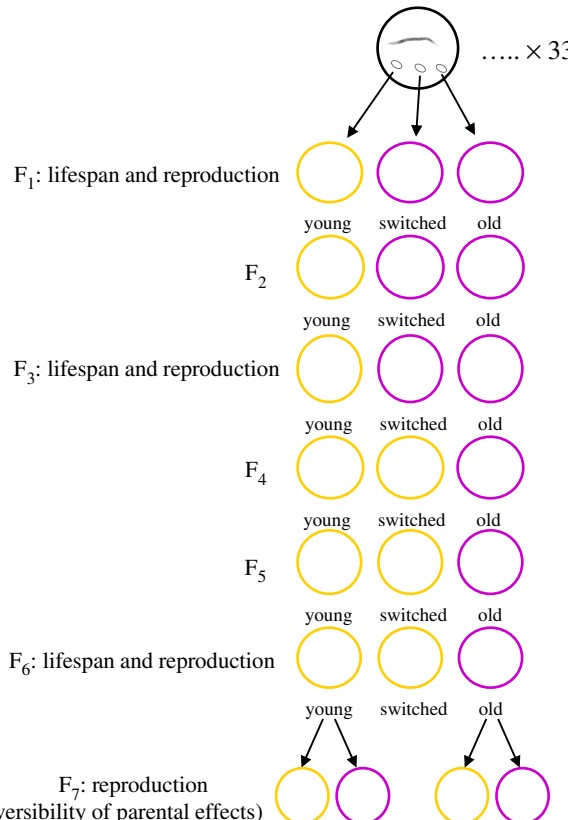

**Figure 1.** Schematic of accumulation and reversibility of parental effects assays: offspring from 1-day old worms (*young parental propagation regime: yellow*) and from 3-day old worms (*old parental propagation regime: purple*) were propagated for six consecutive generations. In the additional parental regime (*switched*), offspring from 3-day old adults were propagated for the first three generations, followed by three generations of offspring from 1-day old adults. Lifespan and reproduction were measured in offspring from the three parental regimes in generations one, three and six. In generation seven, reproduction was measured in offspring from 1- and 3-day old adults from both the *young* and the *old parental propagation regimes*. (Online version in colour.)

propagation regime, so that parental age at egg laying was kept constant in each generation. To test whether parental effects on lifespan and fitness were accumulating and amplifying over successive generations, we measured lifespan and reproduction in offspring from the three regimes in generation one, three and six (see below for details).

## (c) Reversibility of accumulated parental effects assay

To test whether accumulated parental effects on fitness were reversible after six generations, we continued the experiment for an additional generation (generation seven). In this final generation, we used a fully factorial design to measure reproduction in offspring from 1- and 3-day old adults from both the *young* and the *old* propagation regimes (figure 1).

## (d) Reproduction and lifespan assays

In the reproduction assays conducted at generation one, three, six and seven, we placed single unmated (i.e. self-fertilized) hermaphrodites on individual plates from the first day of adulthood until day 5 (the reproductively active days). We moved worms onto new plates every 24 h, and incubated the eggs laid for 48 h at 20°C. After 48 h, we then killed the offspring by heat shock at 40°C and counted the number of offspring.

We used the same parents to measure reproduction and lifespan. We conducted daily mortality checks and after reproduction

ceased we transferred the worms onto new plates every second day until death. Death was defined as the absence of movement in response to touch.

## (e) Egg size, development time and adult size of offspring from young and old parents

In a separate set of assays from the multigenerational experiment, we investigated the effect of parental age on offspring in more detail. We measured the egg size of young (1-day old) and old (3-day old) parents, and to ascertain whether any change in egg size continued even later in life, we also measured eggs laid by 4-day old parents. We also measured developmental time and body size at maturity for offspring from young (1-day old) and old (3-day old) parents.

## (f) Egg size

We measured egg size from 1, 3 and 4-day old parents. In addition to the parental adult ages day 1 (young) and day 3 (old) used in the propagation experiment, we included an additional age category (4-day old adult) to examine egg size in parents at the very end of their reproductive period. To generate parents for the egg size assay, we picked eggs from a 1 h synchronized egg laying of 2-day old adults and placed each egg on an individual plate. We allowed the eggs to develop into adults and moved all worms onto new plates at day 1, 3 and 4 of adulthood. After the transfer to a new plate, each worm was continually observed for the presence of newly laid eggs for 4.5 h. We collected two eggs from each of the 1- and 3-day old parents and measured the size of the egg immediately. Owing to low reproduction at adult day 4, most of the worms did not reproduce at 4 days old. To maximize the number of eggs, where possible, we collected two eggs from each parent that reproduced during this period. We placed the eggs on top of a thin 2% agarose pad [42] that had not been allowed to dry completely, on top of an objective glass. We placed a drop of M9 salt solution on top of each egg to clearly visualize the eggs. We used a Leica M165C microscope set to 120× magnification. The surface area of each egg was measured by taking photographs of the newly laid eggs using a Lumenera Infinity 2-5C digital microscope camera, and the photos were analysed using IMAGEJ [43].

## (g) Development time and adult size

We measured the time to maturity and size at maturity of offspring from 1- and 3-day old parents. For this, we allowed worms to lay eggs for 1 h during their first and third day of adulthood. We picked single eggs onto individual plates. At the day of expected maturation of these offspring, we checked via observation once every hour for the appearance of a fully formed vulva (end of fourth moult) to determine maturation. When each worm was determined to have reached maturity, it was transferred to a new plate for immediate photography.

To measure adult body size, we photographed the matured adult using a Lumenera Infinity 2-5C digital microscope camera, attached to a Leica M165C microscope. We also photographed a stage micrometre for accurate calibration of sizes. We analysed the photos using *WormSizer* [44] in *Fiji* [45]. These data are presented as the surface area of the worms, according to the algorithm used by *WormSizer*.

## (h) Statistical analyses

We used R v. 3.5.1 [46] for all statistical analyses and figures. In analyses from the intergenerational experiment where assays were conducted in generations one, three and six, we included the main effects of parental propagation regime (*young*, *old* and *switched*) and generation, and their interaction. We also fitted generation as a quadratic term and its interaction with parental

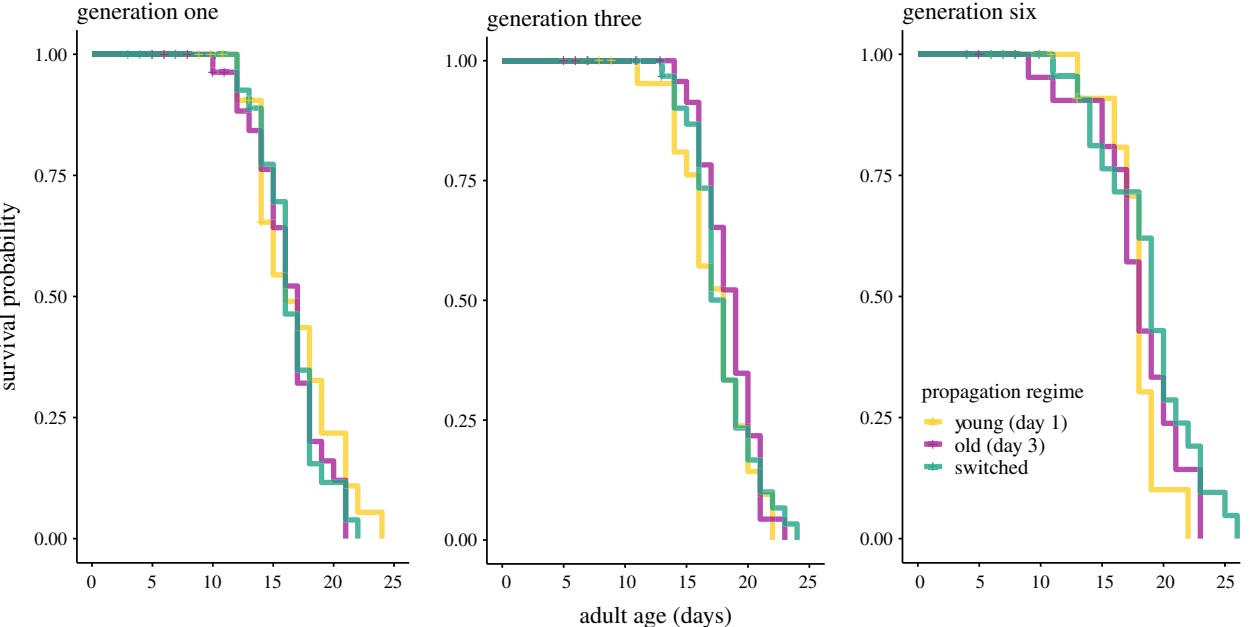

**Figure 2.** No difference in lifespan between parental propagation regimes after one, three or six generations. Survival curves of offspring from *young* (yellow), *old* (purple) and *switched* (green) parental age propagation regimes. (Online version in colour.)

age regime to investigate nonlinear changes across generations. Generation was fitted as a continuous fixed effect in all analyses, except the analyses of pairwise comparisons, where generation was fitted as a factor. The ancestral lines from which the 33 propagation lines originated were fitted as random intercepts. We also fitted random slopes to allow the effect of treatment and generation to vary between ancestral lines. We used stepwise model selection based on likelihood ratio tests to compare nested models starting with testing higher-order terms such as interactions and quadratic terms.

We analysed lifespan in offspring from the three propagation regimes in the first, third and sixth generation using mixed effects Cox proportional hazard models in the *coxme* survival package [47]. We fitted propagation regime and generation as fixed effects along with their interaction, and ancestral lines as a random effect. We removed individuals that died of matricide over the three parental age regimes ($n = 41$ young, $n = 25$ old, and $n = 17$ switched) or escaped from agar ($n = 5$). With increasing number of generations of propagation, some lines were lost by inability to produce offspring during the set time of egg laying (3 h during adult day one or three depending on propagation regime). In total, 196 individuals (young = 50, old = 69, switched = 77) were included in the final lifespan analyses.

We examined differences in total reproduction between the three propagation regimes in generation one, three and six. We also used day-specific reproduction data to calculate rate-sensitive individual fitness $\lambda_{\mathrm{ind}}$ which encompasses when and how many offspring are produced [48,49]. $\lambda_{\mathrm{ind}}$ is estimated by solving the Euler–Lotka equation for each individual using the *lambda* function in the *popbio* package and is analogous to the intrinsic rate of population growth [50,51]. We fitted linear mixed models (LMMs) to total reproduction and log-transformed $\lambda_{\mathrm{ind}}$ using the *lmer* function in the *lme4* package [52]. Finally, we performed *post hoc* pairwise comparisons on total reproduction and $\lambda_{\mathrm{ind}}$ (with generation fitted as a factor) using the *emmeans* package [53]. Across the three propagation regimes and three generations, 275 individuals ($n = 92$ young, 90 old, and 93 *switched*) were included in the analyses of total reproduction, $\lambda_{\mathrm{ind},}$ and pairwise comparisons.

To investigate the reversibility of parental effects in generation seven, we calculated $\lambda_{\mathrm{ind}}$ and fitted an LMM with parental propagation regime (*young* and *old*) and the proximate

parental age (1- and 3-day old parents) included as a fixed effect, and ancestral lines as random intercepts. Between 25 and 30 individuals per propagation regime x parental age combination were included in the analyses ($n = 110$).

To examine differences in egg size at different parental ages, we excluded visibly misshaped eggs from the analyses and included only eggs from individuals that produced at least two normal eggs during both day 1 and day 3. For day 4, all available eggs were analysed. We also included only eggs at the gastrula stage, to avoid possible interactions between egg size and developmental stage. This stage has previously been suggested to be the developmental stage where eggs are typically laid [54]. A total of 283 eggs (day 1 = 133, day 3 = 131, day 4 = 19) across two experimental blocks were included in the analyses using LMMs implemented in the *lme4* package. We fitted parental age (1-, 3- and 4-day old) as a fixed effect, and included random intercepts for blocks and random intercepts and random slopes for parent identity (ID).

Similar to the egg size analyses, we only included development time and adult size from offspring of individuals that produced at least two normal eggs during both days 1 and 3. For development time, we analysed 27 individuals per parental age (1- and 3-day old). For adult size, 47 individuals per parental age were analysed. For development time and adult size, we fitted parental age as a fixed effect, and random intercepts for parent ID.

## 3. Results

### (a) No effect of parental age regime on lifespan across generations

Parental age did not affect offspring lifespan, neither after a single generation nor after allowing time for potential accumulation over multiple generations. Lifespan increased significantly across generations in all three parental age regimes ($\beta = -0.124$, $z = -3.17$, $p = 0.001$; electronic supplementary material, table S1a,b; figure 2). Importantly, there was no significant effect of parental propagation regime ($\chi_2^2 = 0.18$, $p = 0.91$) or its interaction with generation on lifespan

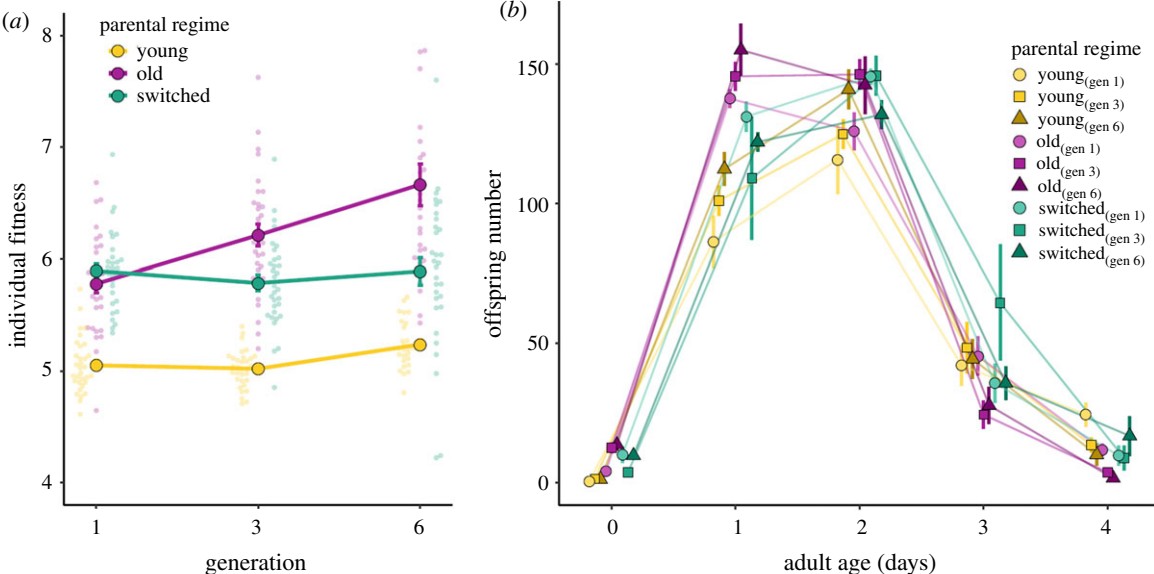

**Figure 3.** (a) Individual fitness for three parental propagation regimes across generations. Individual fitness $\lambda_{ind}$ ($\pm$s.e.) of offspring from *young* (yellow), *old* (purple) and *switched* (green) parental age propagation regime after one, three and six generations. Small points represent raw data. (b) Age-specific reproduction for three parental propagation regimes across generations. Age-specific reproduction ($\pm$s.e.) of offspring from *young* (yellow), *old* (purple) and *switched* (green) parental age propagation regime after one, three and six generations. Generations (1, 3 and 6) are denoted by colour gradients. (Online version in colour.)

($\chi^2_2 = 3.863$, $p = 0.145$). Thus, parental propagation regime had no effect on offspring lifespan.

## (b) Increased fitness in offspring from old parental regime across generations

We found parental age effects on offspring total reproduction and individual fitness ($\lambda_{ind}$) across one generation, although we found some evidence for cumulative effects over successive generations. We found a significant effect of parental propagation regime on total reproduction ($\chi^2 = 63.87$, $p < 0.001$), with worms from the *old* parental regime producing significantly more offspring in total than the *young* parental regime ($\beta = 42.560$, $t_{1,208} = 4.9$, $p < 0.001$; electronic supplementary material, figure S1 and table S2). There was also a significant interaction between propagation regime and generation; total reproduction increased across generations in the *young* $\beta = 4.870$, $t_{1,156} = 2.9$, $p = 0.004$), and (less so) in the *old* parental regime (interaction with *young* $\beta = -2.328$, $t_{1,188} = -1.0$ $p = 0.322$). Like the increase in lifespan across all parental regimes, we also found increased total reproduction in both the young and old parental propagation regimes over successive generations, which may be owing to a general increase in worm condition over time. Total reproduction decreased across generations in the switched regime, probably owing to switching from the old parental propagation in the first three generations to the young propagation regime in generations four to six (interaction with *young* $\beta = -5.809$, $t_{1,184} = -2.537$, $p = 0.012$; electronic supplementary material figure S1 and table S2).

Worms from the *old* propagation regime also had significantly higher fitness ($\lambda_{ind}$) than the *young* regime ($\beta = 0.091$, $t_{1,201} = 3.123$, $p < 0.005$: figure 3a,b; electronic supplementary material, table S2). The higher fitness in the old propagation regime further increased across generations (interaction $\beta = 0.05$, $t_{1,188} = 3.84$, $p < 0.001$; electronic supplementary material, table S2), which shows an accumulation of positive parental age effects in the old parental propagation regime

across generations. See the electronic supplementary material, table S3 for pairwise comparisons of total reproduction and $\lambda_{ind}$ in generations one, three and six.

## (c) Accumulated parental effects are reversible after one generation

In generation seven, we investigated the reversibility of parental age effects that had potentially accumulated over six generations. To do this, we measured total reproduction and fitness ($\lambda_{ind}$) in offspring from young (1-day old) and old (3-day old) parents from both the *young* and *old* parental regimes. Offspring of old parents produced more offspring ($\beta = 25.263$, $t_{1,3.4} = 3.485$, $p < 0.001$; electronic supplementary material, figure S2) and had significantly higher fitness ($\lambda_{ind}$) than offspring of young worms ($\beta = 0.198$, $t_{1,34} = 5.094$, $p < 0.001$), irrespective of parental propagation regime (electronic supplementary material, table S4; figure 4a,b).

## (d) Increased egg size, development rate and adult size in offspring from old parents

Old worms produced larger eggs than young worms ($\chi^2 = 52.293$, $p < 0.001$; electronic supplementary material, table S5). On average, eggs laid by 1-day old parents were 7% smaller than those laid by 3-day old worms, while eggs laid on day 4 were 6% larger than 3-day old parents. The larvae emerging from old individuals also developed significantly faster ($\beta = -0.045$, $t_{1,26} = 2.88$, $p < 0.05$) and when measured at sexual maturity, the resulting offspring were significantly larger than offspring of young worms ($\beta = 0.005$, $t_{1,46} = 5.72$, $p < 0.001$; electronic supplementary material, table S5; figure 5).

## 4. Discussion

In this study, we found no reduction in lifespan in offspring from older parents. Contrary to detrimental effects on offspring

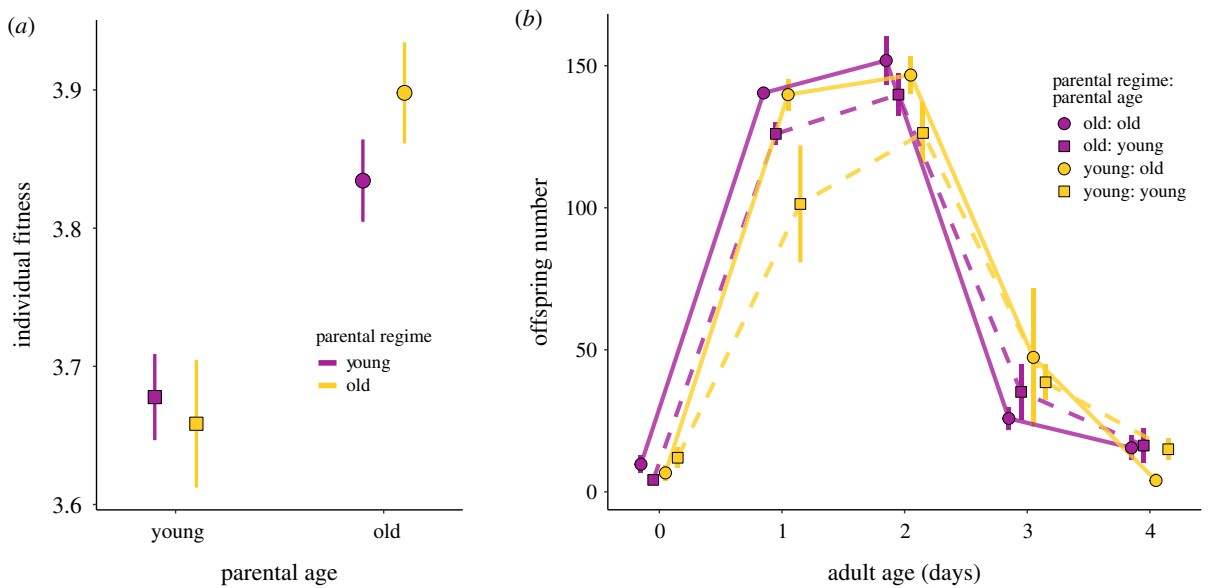

**Figure 4.** (a) Reversibility of parental age effects in generation seven. Individual fitness $\lambda_{ind}$ (±s.e.) of offspring from 1- and 3-day old parents (proximate parental age on the X-axis) generated from the *young* (yellow) and *old* (purple) parental propagation regime at generation seven. (b) Age-specific reproduction in generation seven. Age-specific reproduction (±s.e.) of offspring from 1- (squares) and 3-day (circles) old parents (proximate parental age) generated from the *young* (yellow) and *old* (purple) parental propagation regime at generation seven. (Online version in colour.)

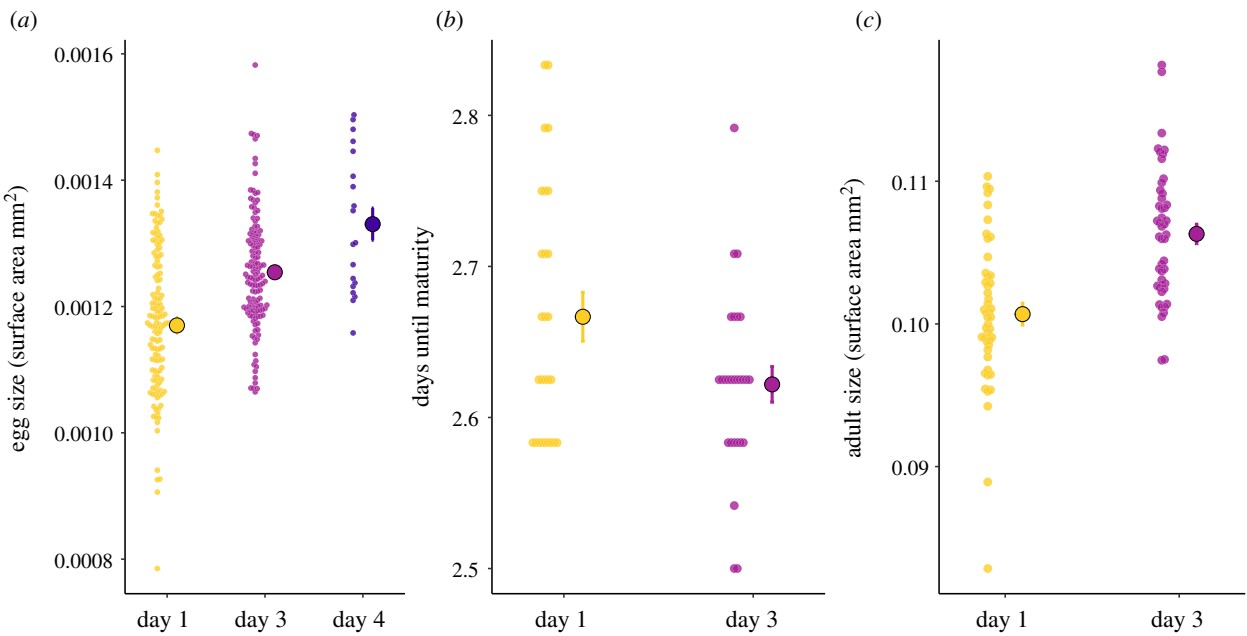

**Figure 5.** Egg size, development time and adult size of offspring from young, old and very old parents. (a) Egg size: egg size of offspring from 1-, 3- and 4-day old parents. (b) Developmental time: number of days from egg laying until sexual maturity in offspring from 1- and 3-day old parents. (c) Adult size: size of offspring at sexual maturity from 1- and 3-day old parents. Small points represent raw data. (Online version in colour.)

phenotype, we found that offspring of old parents had higher fitness than offspring of young parents. Specifically, our results show older parents lay larger eggs and develop faster into larger adults that have increased early-life reproduction. This fitness advantage to offspring of older parents manifested after a single generation of breeding and was maintained throughout six generations of selection. Crucially, the beneficial effect of old parental age on offspring fitness increased with the number of generations. The effects were fully reversed following a single generation of breeding from a young parent.

A few recent studies have investigated the effect of parental age on offspring performance across multiple generations. Qazi *et al.* [55] found negative effects of maternal age

on embryo viability and larval survival over two generations in fruit flies. However, there was a complex interaction between maternal age and the age of the grandmother at the time of conception of the mother, with effects also depending on strain. Grandmaternal effects on egg hatching have also been reported in *Drosophila serrata* [25]. Wylde *et al.* [8] also reported an interaction between maternal and paternal age effects on offspring lifespan over two generations in neriid flies *Telostylinus angusticollis*. More than seven decades ago, multi-generation experiments led Lansing to suggest that old parents pass on an 'ageing factor' to their offspring that accumulates across generations [12,13]. Our results refute the universality of this hypothesis. Instead,

our findings support the hypothesis that offspring of old parents receive more resources, which in turn bestows them with higher fitness. Crucially, this beneficial parental effect increases across successive generations, in direct opposition to the Lansing effect. Our findings are consistent with a recent study showing that offspring of younger *C. elegans* worms developed slower, into smaller larvae, and had fewer offspring [56]. Moreover, that study found an age-related increase in vitellogenin VIT-2 in embryos suggesting that the phenotypic changes in offspring observed with increased parental age are owing to increased yolk provisioning to embryos by older parents. These findings, in addition to our result of an age-related increase in egg size, suggest that *C. elegans* may increase parental investment with age by packing eggs with increased volumes of yolk that cumulatively increase offspring fitness across successive generations.

Increased early-life reproduction and accelerated senescence in offspring of older parents have been shown in several studies [17,18,34]. This has led to the suggestion that the Lansing effect can result in a phenotypically plastic switch to a 'live fast die young' life-history strategy characterized by an earlier reproductive peak and faster senescence [57]. A key question for evolutionary theory is whether life-history alterations in offspring of old parents are adaptive. In the wild, *C. elegans* live in high-density, colony-like clonal populations [20,58,59]. In this boom-and-bust population cycle, offspring produced early in life are likely to be particularly important for fitness [60]. Therefore, it is very possible that increased investment in fast development and/or early reproduction is adaptive. Our results show that older parents produce eggs that develop faster. Our measure of rate-sensitive fitness ($\lambda_{ind}$), which considers the timing of reproduction, also shows that offspring of older mothers have higher fitness because of reaching their reproductive peak earlier than offspring of young mothers. Similarly, Plaistow *et al.* [61] found that soil mites produce larger offspring late in life in order to increase their competitiveness to access depleting resources. Moreover, in environments where food resources are diminishing owing to a growing population, at the stage when later offspring are born, earlier-produced offspring will have reached an older age and may have an age-related competitive advantage [61]. An analogous finding was made in a study on collared flycatchers that altered the quantity of offspring provisioning not by varying parental age, but by experimental manipulation of brood size. Females raised in smaller broods with lower competition for resources between offspring had increased reproduction in early life and aged faster than females raised in artificially increased broods where competition amongst offspring was higher [62]. Increasing provisioning with age to produce larger offspring that develop faster may be an adaptive strategy to allow later-born offspring to compete with older siblings, thus maximizing the total number of surviving offspring. In line with this argument, offspring produced by older females were found to be better larval competitors at high density than offspring laid by younger females in the housefly *Musca domestica* [63]. Given that the life-history of many invertebrates is typified by large fluctuations in population density, accelerated life-histories in the offspring of older parents may be a general adaptive strategy across taxa.

An organism's reproductive schedule, such as how many offspring of what size it produces at different parental ages, is a crucial aspect of its life-history [50,64,65]. It is predicted that the size and number of offspring should co-vary negatively [66], and that parental age, a reliable indicator of condition, could influence the magnitude of trade-offs [67]. While old females are often limited in their ability to acquire and store body reserves [68–70], *C. elegans* nematodes increase in body size with age [71,72]. Increasing body size with age is likely to allow worms to increase the amount of resources transferred to offspring, which leads to the production of larger eggs and more provisioning to later-born offspring [56].

Why older worms produce fewer offspring could be owing to a constraint on sperm supply late in the reproductive period. Self-fertilizing hermaphrodites produce a finite number of sperm at the last larvae stage (L4) which they use to fertilize ova in adulthood [35]. Cross-fertilization can increase brood size [73], demonstrating that selfing (i.e. non-mated) hermaphrodites are sperm-limited. Therefore, limited in the number of eggs they can fertilize because of low sperm counts, older hermaphrodites' only option to maximise fitness may be to invest more resources into each individual egg. Insufficient sperm supply in older *C. elegans* hermaphrodites could also be owing to a trade-off between sperm production and some other trait important for fitness. Evidence suggests a fitness trade-off between sperm production and development time, with faster larval development favouring the evolution of lower sperm counts and slower larval development favouring the evolution of higher sperm counts [73–76]. By manipulating the duration of larval development via temperature, experimental evolution in *C. elegans* showed that selection for faster larval development did indeed favour the evolution of fewer sperm and vice versa [74]. Additionally, a mutation that increases sperm production by 50% has been found to increase brood size but at a cost of delayed onset of oogenesis and fertilization, resulting in slower development and increased egg-to-egg generation time that negatively affects fitness [73]. These findings highlight the importance of early reproduction for fitness in this species, and the number of sperm produced appears to be part of an adaptive strategy to balance the fitness costs of delayed development.

However, questions have been raised about whether hermaphrodites are in fact sperm-limited. Murray & Cutter [74] found that hermaphrodites reared at high and low temperatures produced more sperm than were used to fertilize all oocytes, suggesting that hermaphrodites were oocyte-limited. A study by Goranson *et al.* [77] also found that *C. elegans* reproduction was not sperm-limited when worms were kept in soil and compost. Thus, while it has previously been found that *C. elegans* are sperm-limited in benign laboratory conditions [35], more recent observations suggest the number of sperm produced is sufficient to fertilize all ova, and that hermaphrodites are oocyte-limited in more realistically challenging environments [74,77,78].

To date, most evolutionary theories of ageing assume that the effects of ageing do not persist to the next generation, i.e. that the fitness or phenotypic quality of offspring is independent of parental age [79–83] (but see [15,27]). However, theoretical interest in how parental age effects influence the evolution of ageing is increasing [4,5,7]. Negative parental age effects on offspring reduce the relative contribution of later-life reproduction to fitness. If offspring produced late in life have a relatively lower likelihood of surviving and reproducing, selection on early-life reproduction will be relatively increased [5,7]. This is predicted to lead to a steeper

age-related decline in the strength of natural selection. The opposite is true for positive parental effects, which would lead to increases in the relative value of late-life reproduction and a shallower decline in the strength of selection with age. A recent model [4] predicted that (i) increasingly beneficial maternal effect on offspring quality may evolve when fertility increases with age faster than the decline in survival probability, in particular early in life, and (ii) selection on maternal effects will decline faster than selection on fertility. Our results are in line with the first prediction because *C. elegans* fertility increases early in life, while survival is high and stable. However, our results contradict the second prediction because parental effects continue to be beneficial at ages when fertility declines. It is possible that the ecology of *C. elegans* leads to strong selection on offspring quality at the cost of offspring number in late life because late offspring need to develop faster to compete for dwindling resources. Overall, these results support the theoretical conjecture that age-specific changes in fertility and offspring quality can diverge and contribute to variation in age-specific life-histories. Given the importance of parental age effects for life-history evolution in general [84] and the evolution of ageing in particular [4,15,27] we need more studies from organisms with diverse life-histories.

Data accessibility. Data are available from the Dryad Digital Repository: https://doi.org/10.5061/dryad.qrfj6q5gz [85].

Authors' contributions. L.M.T.: conceptualization, formal analysis, visualization, writing—original draft, writing—review and editing; H.C.: conceptualization, data curation, methodology, writing—review and editing; M.I.L.: conceptualization, formal analysis, supervision, visualization, writing—review and editing; A.A.M.: conceptualization, funding acquisition, investigation, methodology, resources, supervision, writing—review and editing. All authors gave final approval for publication and agreed to be held accountable for the work performed therein.

Competing interests. The authors declare no competing interests.

Funding. This work was supported by ERC Consolidator Grant GermlineAgeingSoma to A.A.M. and the Swedish Research Council (Vetenskapsrådet, grant no. 2016-05195) to M.I.L.

Acknowledgements. We thank Andreas Sutter for assistance with statistical analyses, and Hwei-yen Chen for helpful feedback on the manuscript.

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
