## [Peer Review File · Proceedings of the Royal Society B: Biological Sciences]

Review History

RSPB-2021-0775.R0 (Original submission)

Review form: Reviewer 1

Recommendation

Major revision is needed (please make suggestions in comments)

Scientific importance: Is the manuscript an original and important contribution to its field?

Good

General interest: Is the paper of sufficient general interest?

Good

Quality of the paper: Is the overall quality of the paper suitable?

Good

Is the length of the paper justified?

Yes

Should the paper be seen by a specialist statistical reviewer?

No

Do you have any concerns about statistical analyses in this paper? If so, please specify them explicitly in your report.

No

It is a condition of publication that authors make their supporting data, code and materials available - either as supplementary material or hosted in an external repository. Please rate, if applicable, the supporting data on the following criteria.

Is it accessible?

Yes

Is it clear?

No

Is it adequate?

No

Do you have any ethical concerns with this paper?

No

Comments to the Author

This is an interesting study of the relationship between aging and fitness in *C. elegans*. It addresses two questions: how parental age affects offspring fitness, and whether such effects can accumulate across generations. Parental effects can be positive (maturation benefits) or negative (senescence effects); transgenerational accumulation of the latter is the Lansing effect. Or they can be both, as where progeny show increased or earlier reproduction coupled to more rapid aging. Using careful experimental protocols and well-designed statistics, the authors detect greater fitness in progeny of older worms, and also an accumulation of such benefits across generations. The findings are of interest in terms *C. elegans* biology and life history evolution (increases in progeny fitness with parental age could have an effect on the evolution of aging), and in the context of recent renewed interest in the Lansing effect. They describe a sort of reverse Lansing effect, presumably resulting from adult maturational changes (perhaps yolk provisioning of embryos, and changes in oogenetic mechanisms leading to increased egg size). These results are novel and intriguing.

While I would support publication of this study in some form, the manuscript needs to be better presented in terms of clarity of argument, and of presentation of the results, which at present are difficult to understand in places due to an overly terse style of presentation. In its present form many readers, particularly from the *C. elegans* lifespan genetics field, will struggle to understand this manuscript (particularly the results section), limiting its impact and utility. Another lesser weakness is that the scope of the study is a little limited in terms of follow up of the findings obtained.

Major points

1) The various phenomena and theories discussed in this study could be presented with a higher degree of logical clarity and precision. The study deals with several distinct claims and questions. (1) Progeny fitness can increase or decrease with parental age, depending on context (e.g. species, environmental conditions). A simple interpretation here is that adult maturation can increase progeny fitness while adult senescence decreases progeny fitness. This underscores the two sides of adult aging: positive (maturation) and negative (senescence). This could be made clearer. The presence of these opposite effects is not a contradiction. (2) The Lansing effect refers specifically to intergenerational deterioration of successive progeny of old parents; Lansing suggested the hypothesis of an accumulating deteriorative factor, but this hypothesis is distinct from the Lansing effect. At times in the manuscript, there is insufficiently clear distinction between (1) and

(2). This affect the introduction and discussion, and also the results section.

2) Line 307-308 in the Results. This is unclear. The issue here, surely, is whether large eggs give rise to large worms. As I understand it, this is saying that large eggs give rise to worms that are larger than eggs from young worms. But that could be because progeny of young worms are smaller, not because the eggs of young worms are smaller. What is needed is a comparison of egg size vs worm size using eggs from old worms. This needs to be included in the study, performing additional tests if necessary. It would also be interesting to know whether other fitness parameters were increased in worms hatched from larger eggs, e.g. earlier reproductive schedule and increased overall fertility. How does egg size covary with fitness traits: is it the bigger the egg, the fitter the worm?

Minor points

Line 28. "Such parental effect senescence is" is slightly unclear. "Such parental senescence effects are" ?

Line 29 "higher quality" ?

Line 36. The reasoning in the last two sentences of the abstract is inexact. "contradict the theory" could imply that this finding argues that the Lansing effect theory is incorrect. But the Lansing effect is an effect (not a theory) that is seen in various organisms (e.g. rotifers). More appropriate would be to make the final sentence the penultimate, changing to "that in *C. elegans* offspring of" (to avoid the misapprehension that this is a general claim) and the penultimate the final, remarking that this is the opposite of the Lansing effect (perhaps even labelling it a "reverse Lansing effect" or something similar.

Introduction

Line 44. "many taxa and is the subject of renewed theoretical and empirical interest" Here "renewed" would help the reader to understand that the Lansing effect is a topic that has lain relatively dormant for a long time, but has recently reawakened.

Line 46. "suggested" or "found" ? A point of emphasis: it would be more surprising if offspring quality did not decline with age, given the presence of parental senescence. What is surprising about the Lansing effect is the inter-generational effect.

Line 54. As written it appears that the Lansing effect as a phenomenon is here being conflated with Lansing's theory about the mechanism underlying the effect. In a similar way, rate-of-living effects on lifespan do occur, but the rate-of-living theory that rate-of-living effects are attributable to metabolic rate is incorrect. The authors need to take care to distinguish the Lansing effect from Lansing's hypothesis about the mechanism involved.

Line 56. The meaning of a decline in developmental time is unclear; would that not be an increase in fitness? "developmental rate" ?

Line 57-58. Has anyone ever tested whether there is a Lansing effect in women? I am guessing not, and perhaps worth mentioning that it is not know whether such an effect exists. Risks of some aneuploidies increase with age and can be passed to subsequent generations, e.g. Down's trisomy 21. Would that count as a Lansing effect?

Line 63-64. "may instead" Again, important to emphasize that what happens depends on context (species, environment?). "documented. However, in some contexts older parents" which would mean removing "Indeed" from line 66.

Line 78-79. Arguably this will depend on whether maturational or senescent facets of aging predominate. I think this would be worth arguing here.

Line 79-87. The arguments about evolutionary theory are cogent and apposite.

Line 88-90. "understand", better "explore"? It would be more precise to separate topics (1) and (2) as described above, rather than conflating them.

Line 90. "from young or old parents." ?

Line 95. Better "mating with males" ? "cross-fertilization" could imply genotypic differences between the sexes playing a role here, but the two sexes are generally isogenic.

Line 106. "cumulative inter-generational effects" ?

Line 114. "wild-type"

Line 115-116. "combine maternal and paternal effects". This is rather inexact. Hermaphrodites are really self-fertilizing females. Aging (maturation, senescence) will be affecting oogenesis, and the physiology that supports it, but not spermatogenesis. The male part is represented only by the presence of the sperm, generated earlier. Thus, paternal effects are only represented to a lesser extent. A question is whether sperm competition leaves the worst sperm to last (e.g. nullo X sperm? Male self progeny appear more often among the last progeny, if I recall correctly).

Line 117. "cohorts". "stocks" ?

Line 119. "adults in each"

Line 121-122. ampicillin, streptomycin, nystatin without capitals. These are not trade names.

Line 126. "0.2 ml E. coli seeding". Rephrase to make this more exact (seeding with an E. coli suspension?).

Line 156. "haphazardly". "randomly"?

Line 158. "egg layings in each generation" ?

Line 160. "magnifying." "amplifying"?

Line 202. "12x" "120x"?

Line 233. "cox proportional hazard", "Cox" (after David Cox)

Results

I found the whole results section too terse and telegraphic. Inclusion of more exposition of the meaning of the results would improve clarity and readability of this section. Help the reader to understand what the results mean. It would also help if the subsection headings more clearly indicated what the data is showing, rather than what is being looked at.

Line 273. The finding that lifespan increased across generations across all parental age regimes is unexpected, and some sort of remark here is warranted to avoid the bafflement of the reader, even if only "for reasons that remain unclear" or "for reasons discussed below". But this results does not seem to be discussed anywhere (or have I missed it?). It needs to be discussed.

Line 279. I found this whole section presented in a confusing way. It needs to be made much clearer what exactly the results are saying and what they mean.

Line 280-282. Here again the difference between (1) and (2) should be spelled out better. I take it that "proximal effects" refers to (1), but this wants to be clearer.

The section overall seems to say, first: that later progeny are fitter, a claim relating to (1). Then, relating to (2), that intergenerational increases in fitness are seen in both the young and old groups, arguing against a transgenerational accumulation of increased fitness from older parents, but this result, like the increase in lifespan, is odd and unexpected: why is everybody getting fitter? But then it is stated that, after all, fitness is increased transgenerationally more in the old group. I take it that the point here is that two measures of fitness are made, and one but not the other indicates transgenerational accumulation of increased fitness from older parents. The results section needs to include at least indication of the conclusions that can be drawn from these findings, and how they can be drawn. How is it that there is not an old parent transgenerational benefit with respect to reproductive fitness, but for fitness there is?

Line 300-303. Again, the conclusions to be drawn from this section are unclear. The question is surely, if you compare early progeny of early- and late-propagated worms, is the greater fitness of the latter still there? Clarity needs to be improved.

Line 305. It would be helpful here to describe the change in size, in terms of percentage change in length and volume, and how the increases in size vary (average change), the size of the biggest eggs seen. It would be good to include an image showing a normal size egg and a giant egg.

Line 305-306. Unclear. I think this means that the larvae emerging from larger eggs develop more quickly, rather than that the eggs develop more quickly within the hermaphrodite gonad.

Line 307. Remove . (period)

Discussion

Line 310. "crucial", hyperbole. "important" ?

Line 311. "and has ramifications in", "and is relevant to" ?

Line 316. Is that really "with" or should it in fact be "and"? Data should be presented to clearly establish the "with" (see major point 2 above).

Line 318-319. "and was maintained throughout six generations" This is ambiguous, and could give the impression that fitness benefits in progeny of old worms were present for the subsequent 6 generations regardless of parental age. "and continued to accumulate over six generations of selection" (unless I have misunderstood something).

Line 330-331. This is not a cogent conclusion; see my comments above (e.g. on line 36).

Line 335. "are further supported by", better, "are consistent with"

Line 337. VIT-2 is a component of YP170.

Line 339-341. The Perez et al study clearly implies this, but it is not made clear here how the present study does. Surely the increase in egg size suggest this, and this would be worth proposing here: the increase in egg size could imply better provisioning, but it doesn't exactly prove this. Possibly the volume of the embryo is increased to accommodate larger numbers of yolk and lipid droplets, supporting faster growth within the egg, or perhaps resulting in

emergence of a larger L1 from the egg, with a head start in growth terms.

Line 341. It would be interesting to measure vitellogenin levels in mothers on day 1-4, and in eggs, in successive generations.

Line 343-344. "the Lansing effect is a result". Surely "Lansing effects can result"

Line 349. "it is conceivable", surely "it is very possible", particularly given the Perez et al study, among others.

Line 353. The argument from here seems highly plausible. It also begs the question: do eggs from older adults (or larger eggs) give rise to more starvation resistant larvae? Have earlier studies tested this (at least for eggs from older adults)?

Line 375. Larger eggs yes, but fewer? Or is this referring to number of eggs laid later by worms arising from later eggs?

Line 380-381. Strange statement. The cessation of egg laying is caused by self-sperm depletion. (Possibly this sentence is intended to express a different idea).

Line 382. "all ova". This is not true for lab conditions where many ova remain unfertilized at the point of sperm depletion; while this remark is justified in the light of later discussion about wild conditions, qualification here would be appropriate, to avoid confusing the reader.

Line 383. "suggesting", surely "demonstrating"

Line 384-386. Plausible argument.

Line 397. "may be part" "appears to be part"?

Supplement

Raw data is provided in 6 Numbers files, but a description of the contents of these files needs to be presented (e.g. in the main Supplementary Materials file).

One raw data file includes raw data for survival analysis which, if a contents description is provided, which (as other raw data files) is helpful for anyone wishing to reanalyze the data. But for the ordinary reader, a table of basic lifespan related statistics should be included in the usual fashion (showing number of trials, sample sizes, numbers of censored values, mean lifespan and appropriate statistical comparisons).

Review form: Reviewer 2

Recommendation

Accept with minor revision (please list in comments)

Scientific importance: Is the manuscript an original and important contribution to its field?

Excellent

General interest: Is the paper of sufficient general interest?

Excellent

Quality of the paper: Is the overall quality of the paper suitable?

Good

Is the length of the paper justified?

No

Should the paper be seen by a specialist statistical reviewer?

No

Do you have any concerns about statistical analyses in this paper? If so, please specify them explicitly in your report.

No

It is a condition of publication that authors make their supporting data, code and materials available - either as supplementary material or hosted in an external repository. Please rate, if applicable, the supporting data on the following criteria.

Is it accessible?

No

Is it clear?

No

Is it adequate?

No

Do you have any ethical concerns with this paper?

No

Comments to the Author

This is an elegant study. Well designed and adequately replicated. However, I thought the Introduction could be more focused and the references better distributed to support the relevant elements/statements (see below). Overall, there are perhaps, too many references. One reason this seems to arise is because of the Discussion spends too long speculating about the specific limitation of the amount of sperm available in this hermaphrodite.

Ln 43 The first sentence with it's eleven references at the end is not very helpful. It would be better if the relevant references appeared after "reported in many taxa" and separated from those relevant to each of "much empirical" and "theoretical interest". On what basis did you chose the examples in many taxa?

Ln 52 Insert "further work on pre-industrial " before "humans"

Ln 52 start a new paragraph at "The lifespan..."

Ln54-57. Unbrigade the references here and distribute them to the relevant trait!

Ln 57 we are back to humans. And is this not a mechanism? So why is the very next sentence beginning "Several mechanisms, So this new paragraph I'm suggesting goes Humans, General, Human, General, I suggest you could have a much better, shorter opening paragraph with a better flow. The subsequent paragraphs Ln 63-77 and Ln 78-87 are much better and then quickly get us to what you did Ln 88.

Ln 101-105 and 105-107 is repeated later in the Methods. Is it needed here?

Ln 156 "haphazardly" do you mean "randomly"?

Ln 184 why was "very old (four day old) parents" added here for the first time?

Ln 236-240. Are the rates of matricide significantly different between the parental age regimes?

Ln 277 Make it clear both these Chi-sq are for 2 degrees of freedom

Ln 288 I think it would be better to present if old parental slope 2.532 is significant rather than theres no significant difference from young.

Ln 309 - Discussion I would suggest you start with a Summary of your findings. So begin this section from Ln 313 - "In this study" remove the redundant ", " after "study". The first two

sentences of the current Discussion can be used later.

Decision letter (RSPB-2021-0775.R0)

10-May-2021

Dear Dr Travers:

I am writing to inform you that your manuscript RSPB-2021-0775 entitled "Beneficial cumulative effects of old parental age on offspring fitness" has, in its current form, been rejected for publication in Proceedings B.

This action has been taken on the advice of referees, who have recommended that substantial revisions are necessary. With this in mind we would be happy to consider a resubmission, provided the comments of the referees are fully addressed. However please note that this is not a provisional acceptance.

Sincerely,
Dr Maurine Neiman
<mailto:proceedingsb@royalsociety.org>

Associate Editor
Board Member: 1
Comments to Author:

In this study, the authors conduct a multigenerational experiment to test the effects of parental age on offspring fitness in nematodes; presenting results that are at odds with those predicted by the often-observed Lansing effect (the observation offspring of old parents have lower fitness in certain taxa). Rather, here the authors find that offspring born to older parents have higher fitness, and who reach their reproductive peak earlier. The effects increase in strength over

multiple generations of breeding of old age parents, but are reversible with switching to one generation of breeding by a younger parent.

The paper was reviewed by two referees, each of whom agrees that this study has produced some fascinating results. But each referee has indicated that major revisions are required.

Referee 1 provided a very thorough review, with many comments that require careful attention. They noted a major rewrite is required for conceptual precision, and to make the results clearer (particularly for readers in the *C. elegans* ageing field). This will require a clearer narrative through the results, which the referee notes as quite terse and disjointed. The referee also questions the results of the egg size data, since the data is potentially confounded with the body size data of the young and old parents. Disentangling potentially confounding effects of egg and parent body size may require the authors to conduct a supplemental experiment, unless the data are already collected; or otherwise to modify their inferences.

Referee 2 has noted the Introduction and Discussion are overly long, with too many citations (for example, the first sentence is followed by 11 citations). They have also provided suggestions / queries of some of the statistical analyses.

On that note, I also have a query on the statistical approach, since I believe it is likely that the mixed effects models are mis-specified – in terms of the level of replication at which the parameter estimates are modelled. The denominator degrees of freedom seem to indicate that the data are modelled at the level of the individual; but I believe the appropriate level to model the data is the level of the ‘ancestral line’ – there are 33 ancestral lines; lest the analyses be pseudoreplicated. Generally, adding a random intercept for “Line ID” will suffice when it comes to estimation of main fixed effects in a linear mixed model, but the interactions between fixed effects are still likely to be estimated at the incorrect (individual) level, leading to inherent pseudoreplication in the models. Correcting for this would require addition of random slopes (e.g. 1 + propagation regime + generation + propagation regime:generation | ancestral.line). This same issue would apply for all analyses (including the survival models, coxme, and including analyses on reversibility of parental effects). For further information on this issue, please see:

Arnqvist, G. (2020). "Mixed Models Offer No Freedom from Degrees of Freedom." *Trends in Ecology & Evolution*.

Reviewer(s)' Comments to Author:

Referee: 1

Comments to the Author(s)

This is an interesting study of the relationship between aging and fitness in *C. elegans*. It addresses two questions: how parental age affects offspring fitness, and whether such effects can accumulate across generations. Parental effects can be positive (maturation benefits) or negative (senescence effects); transgenerational accumulation of the latter is the Lansing effect. Or they can be both, as where progeny show increased or earlier reproduction coupled to more rapid aging. Using careful experimental protocols and well-designed statistics, the authors detect greater fitness in progeny of older worms, and also an accumulation of such benefits across generations. The findings are of interest in terms *C. elegans* biology and life history evolution (increases in progeny fitness with parental age could have an effect on the evolution of aging), and in the context of recent renewed interest in the Lansing effect. They describe a sort of reverse Lansing effect, presumably resulting from adult maturational changes (perhaps yolk provisioning of embryos, and changes in oogenetic mechanisms leading to increased egg size). These results are novel and intriguing.

While I would support publication of this study in some form, the manuscript needs to be better presented in terms of clarity of argument, and of presentation of the results, which at present are difficult to understand in places due to an overly terse style of presentation. In its present form

many readers, particularly from the *C. elegans* lifespan genetics field, will struggle to understand this manuscript (particularly the results section), limiting its impact and utility. Another lesser weakness is that the scope of the study is a little limited in terms of follow up of the findings obtained.

Major points

1) The various phenomena and theories discussed in this study could be presented with a higher degree of logical clarity and precision. The study deals with several distinct claims and questions. (1) Progeny fitness can increase or decrease with parental age, depending on context (e.g. species, environmental conditions). A simple interpretation here is that adult maturation can increase progeny fitness while adult senescence decreases progeny fitness. This underscores the two sides of adult aging: positive (maturation) and negative (senescence). This could be made clearer. The presence of these opposite effects is not a contradiction. (2) The Lansing effect refers specifically to intergenerational deterioration of successive progeny of old parents; Lansing suggested the hypothesis of an accumulating deteriorative factor, but this hypothesis is distinct from the Lansing effect. At times in the manuscript, there is insufficiently clear distinction between (1) and (2). This affects the introduction and discussion, and also the results section.

2) Line 307-308 in the Results. This is unclear. The issue here, surely, is whether large eggs give rise to large worms. As I understand it, this is saying that large eggs give rise to worms that are larger than eggs from young worms. But that could be because progeny of young worms are smaller, not because the eggs of young worms are smaller. What is needed is a comparison of egg size vs worm size using eggs from old worms. This needs to be included in the study, performing additional tests if necessary. It would also be interesting to know whether other fitness parameters were increased in worms hatched from larger eggs, e.g. earlier reproductive schedule and increased overall fertility. How does egg size covary with fitness traits: is it the bigger the egg, the fitter the worm?

Minor points

Line 28. "Such parental effect senescence is" is slightly unclear. "Such parental senescence effects are" ?

Line 29 "higher quality" ?

Line 36. The reasoning in the last two sentences of the abstract is inexact. "contradict the theory" could imply that this finding argues that the Lansing effect theory is incorrect. But the Lansing effect is an effect (not a theory) that is seen in various organisms (e.g. rotifers). More appropriate would be to make the final sentence the penultimate, changing to "that in *C. elegans* offspring of" (to avoid the misapprehension that this is a general claim) and the penultimate the final, remarking that this is the opposite of the Lansing effect (perhaps even labelling it a "reverse Lansing effect" or something similar).

Introduction

Line 44. "many taxa and is the subject of renewed theoretical and empirical interest" Here "renewed" would help the reader to understand that the Lansing effect is a topic that has lain relatively dormant for a long time, but has recently reawakened.

Line 46. "suggested" or "found" ? A point of emphasis: it would be more surprising if offspring quality did not decline with age, given the presence of parental senescence. What is surprising about the Lansing effect is the inter-generational effect.

Line 54. As written it appears that the Lansing effect as a phenomenon is here being conflated with Lansing's theory about the mechanism underlying the effect. In a similar way, rate-of-living

effects on lifespan do occur, but the rate-of-living theory that rate-of-living effects are attributable to metabolic rate is incorrect. The authors need to take care to distinguish the Lansing effect from Lansing's hypothesis about the mechanism involved.

Line 56. The meaning of a decline in developmental time is unclear; would that not be an increase in fitness? "developmental rate" ?

Line 57-58. Has anyone ever tested whether there is a Lansing effect in women? I am guessing not, and perhaps worth mentioning that it is not known whether such an effect exists. Risks of some aneuploidies increase with age and can be passed to subsequent generations, e.g. Down's trisomy 21. Would that count as a Lansing effect?

Line 63-64. "may instead" Again, important to emphasize that what happens depends on context (species, environment?). "documented. However, in some contexts older parents" which would mean removing "Indeed" from line 66.

Line 78-79. Arguably this will depend on whether maturational or senescent facets of aging predominate. I think this would be worth arguing here.

Line 79-87. The arguments about evolutionary theory are cogent and apposite.

Line 88-90. "understand", better "explore"? It would be more precise to separate topics (1) and (2) as described above, rather than conflating them.

Line 90. "from young or old parents." ?

Line 95. Better "mating with males" ? "cross-fertilization" could imply genotypic differences between the sexes playing a role here, but the two sexes are generally isogenic.

Line 106. "cumulative inter-generational effects" ?

Line 114. "wild-type"

Line 115-116. "combine maternal and paternal effects". This is rather inexact. Hermaphrodites are really self-fertilizing females. Aging (maturation, senescence) will be affecting oogenesis, and the physiology that supports it, but not spermatogenesis. The male part is represented only by the presence of the sperm, generated earlier. Thus, paternal effects are only represented to a lesser extent. A question is whether sperm competition leaves the worst sperm to last (e.g. null X sperm? Male self progeny appear more often among the last progeny, if I recall correctly).

Line 117. "cohorts". "stocks" ?

Line 119. "adults in each"

Line 121-122. ampicillin, streptomycin, nystatin without capitals. These are not trade names.

Line 126. "0.2 ml E. coli seeding". Rephrase to make this more exact (seeding with an E. coli suspension?).

Line 156. "haphazardly". "randomly"?

Line 158. "egg layings in each generation" ?

Line 160. "magnifying." "amplifying"?

Line 202. "12x" "120x"?

Line 233. "cox proportional hazard", "Cox" (after David Cox)

Results

I found the whole results section too terse and telegraphic. Inclusion of more exposition of the meaning of the results would improve clarity and readability of this section. Help the reader to understand what the results mean. It would also help if the subsection headings more clearly indicated what the data is showing, rather than what is being looked at.

Line 273. The finding that lifespan increased across generations across all parental age regimes is unexpected, and some sort of remark here is warranted to avoid the bafflement of the reader, even if only "for reasons that remain unclear" or "for reasons discussed below". But this results does not seem to be discussed anywhere (or have I missed it?). It needs to be discussed.

Line 279. I found this whole section presented in a confusing way. It needs to be to made much clearer what exactly the results are saying and what they mean.

Line 280-282. Here again the difference between (1) and (2) should be spelled out better. I take it that "proximal effects" refers to (1), but this wants to be clearer.

The section overall seems to say, first: that later progeny are fitter, a claim relating to (1). Then, relating to (2), that intergenerational increases in fitness are seen in both the young and old groups, arguing against a transgenerational accumulation of increased fitness from older parents, but this result, like the increase in lifespan, is odd and unexpected: why is everybody getting fitter? But then it is stated that, after all, fitness is increased transgenerationally more in the old group. I take it that the point here is that two measures of fitness are made, and one but not the other indicates transgenerational accumulation of increased fitness from older parents. The results section needs to include at least indication of the conclusions that can be drawn from these findings, and how they can be drawn. How is it that there is not an old parent transgenerational benefit with respect to reproductive fitness, but for fitness there is?

Line 300-303. Again, the conclusions to be drawn from this section are unclear. The question is surely, if you compare early progeny of early- and late-propagated worms, is the greater fitness of the latter still there? Clarity needs to be improved.

Line 305. It would be helpful here to describe the change in size, in terms of percentage change in length and volume, and how the increases in size vary (average change), the size of the biggest eggs seen. It would be good to include an image showing a normal size egg and a giant egg.

Line 305-306. Unclear. I think this means that the larvae emerging from larger eggs develop more quickly, rather than that the eggs develop more quickly within the hermaphrodite gonad.

Line 307. Remove . (period)

Discussion

Line 310. "crucial", hyperbole. "important" ?

Line 311. "and has ramifications in", "and is relevant to" ?

Line 316. Is that really "with" or should it in fact be "and"? Data should be presented to clearly establish the "with" (see major point 2 above).

Line 318-319. "and was maintained throughout six generations" This is ambiguous, and could give the impression that fitness benefits in progeny of old worms were present for the

subsequence 6 generations regardless of parental age. "and continued to accumulate over six generations of selection" (unless I have misunderstood something).

Line 330-331. This is not a cogent conclusion; see my comments above (e.g. on line 36).

Line 335. "are further supported by", better, "are consistent with"

Line 337. VIT-2 is a component of YP170.

Line 339-341. The Perez et al study clearly implies this, but it is not made clear here how the present study does. Surely the increase in eggs size suggest this, and this would be worth proposing here: the increase in egg size could imply better provisioning, but it doesn't exactly prove this. Possibly the volume of the embryo is increased to accommodate larger numbers of yolk and lipid droplets, supporting faster growth within the egg, or perhaps resulting in emergence of a larger L1 from the egg, with a head start in growth terms.

Line 341. It would be interesting to measure vitellogenin levels in mothers on day 1-4, and in eggs, in successive generations.

Line 343-344. "the Lansing effect is a result". Surely "Lansing effects can result"

Line 349. "it is conceivable", surely "it is very possible", particularly given the Perez et al study, among others.

Line 353. The argument from here seems highly plausible. It also begs the question: do eggs from older adults (or larger eggs) give rise to more starvation resistant larvae? Have earlier studies tested this (at least for eggs from older adults)?

Line 375. Larger eggs yes, but fewer? Or is this referring to number of eggs laid later by worms arising from later eggs?

Line 380-381. Strange statement. The cessation of egg laying is caused by self-sperm depletion. (Possibly this sentence is intended to express a different idea).

Line 382. "all ova". This is not true for lab conditions where many ova remain unfertilized at the point of sperm depletion; while this remark is justified in the light of later discussion about wild conditions, qualification here would be appropriate, to avoid confusing the reader.

Line 383. "suggesting", surely "demonstrating"

Line 384-386. Plausible argument.

Line 397. "may be part" "appears to be part"?

Supplement

Raw data is provided in 6 Numbers files, but a description of the contents of these files needs to be presented (e.g. in the main Supplementary Materials file).

One raw data file includes raw data for survival analysis which, if a contents description is provided, which (as other raw data files) is helpful for anyone wishing to reanalyze the data. But for the ordinary reader, a table of basic lifespan related statistics should be included in the usual fashion (showing number of trials, sample sizes, numbers of censored values, mean lifespan and appropriate statistical comparisons).

Referee: 2

Comments to the Author(s)

This is an elegant study. Well designed and adequately replicated. However, I thought the Introduction could be more focused and the references better distributed to support the relevant elements/statements (see below). Overall, there are perhaps, too many references. One reason this seems to arise is because of the Discussion spends too long speculating about the specific limitation of the amount of sperm available in this hermaphrodite.

Ln 43 The first sentence with it's eleven references at the end is not very helpful. It would be better if the relevant references appeared after "reported in many taxa" and separated from those relevant to each of "much empirical" and "theoretical interest". On what basis did you chose the examples in many taxa?

Ln 52 Insert "further work on pre-industrial " before "humans"

Ln 52 start a new paragraph at "The lifespan..."

Ln54-57. Unbrigade the references here and distribute them to the relevant trait!

Ln 57 we are back to humans. And is this not a mechanism? So why is the very next sentence beginning "Several mechanisms, So this new paragraph I'm suggesting goes Humans, General, Human, General, I suggest you could have a much better, shorter opening paragraph with a better flow. The subsequent paragraphs Ln 63-77 and Ln 78-87 are much better and then quickly get us to what you did Ln 88.

Ln 101-105 and 105-107 is repeated later in the Methods. Is it needed here?

Ln 156 "haphazardly" do you mean "randomly"?

Ln 184 why was "very old (four day old) parents" added here for the first time?

Ln 236-240. Are the rates of matricide significantly different between the parental age regimes?

Ln 277 Make it clear both these Chi-sq are for 2 degrees of freedom

Ln 288 I think it would be better to present if old parental slope 2.532 is significant rather than theres no significant difference from young.

Ln 309 - Discussion I would suggest you start with a Summary of your findings. So begin this section from Ln 313 - "In this study" remove the redundant "," after "study". The first two sentences of the current Discussion can be used later.

Author's Response to Decision Letter for (RSPB-2021-0775.R0)

See Appendix A.

RSPB-2021-1843.R0

Review form: Reviewer 2

Recommendation

Accept with minor revision (please list in comments)

Scientific importance: Is the manuscript an original and important contribution to its field?

Excellent

General interest: Is the paper of sufficient general interest?

Excellent

Quality of the paper: Is the overall quality of the paper suitable?

Excellent

Is the length of the paper justified?

Yes

Should the paper be seen by a specialist statistical reviewer?

No

Do you have any concerns about statistical analyses in this paper? If so, please specify them explicitly in your report.

No

It is a condition of publication that authors make their supporting data, code and materials available - either as supplementary material or hosted in an external repository. Please rate, if applicable, the supporting data on the following criteria.

Is it accessible?

Yes

Is it clear?

Yes

Is it adequate?

Yes

Do you have any ethical concerns with this paper?

No

Comments to the Author

The revision has been very thorough and addressed my previous concerns, as well as adopted my suggestions on a better logical flow. The paper now reads well and should be of great interest to a wide audience.

However, I noticed a few trivial grammatical mistakes, and occasional tendency in the results to start speculating about causation, which might be better left to the Discussion.

Minor comments:

Ln 182 delete "to" after continued.

Ln 276/277 An example of a sentence drifting into explanation that I would save until the Discussion.

Ln 283 comma needed after "generation".

Ln 292 comma needed after "generations" - unless the rest of the sentence is deleted because it is better moved to the Discussion.

Decision letter (RSPB-2021-1843.R0)

10-Sep-2021

Dear Dr Travers

I am pleased to inform you that your manuscript RSPB-2021-1843 entitled "Beneficial cumulative effects of old parental age on offspring fitness" has been accepted for publication in Proceedings B.

The referee(s) have recommended publication, but also suggest some minor revisions to your manuscript. Therefore, I invite you to respond to the referee(s)' comments and revise your

manuscript. Because the schedule for publication is very tight, it is a condition of publication that you submit the revised version of your manuscript within 7 days. If you do not think you will be able to meet this date please let us know.

- DNA sequences: Genbank accessions F234391-F234402
- Phylogenetic data: TreeBASE accession number S9123
- Final DNA sequence assembly uploaded as online supplemental material

- Climate data and MaxEnt input files: Dryad doi:10.5521/dryad.12311

[http://datadryad.org/submit?journalID=RSPB&manu=\(Document not available\)](http://datadryad.org/submit?journalID=RSPB&manu=(Document%20not%20available)) which will take you to your unique entry in the Dryad repository. If you have already submitted your data to dryad you can make any necessary revisions to your dataset by following the above link. Please see <https://royalsociety.org/journals/ethics-policies/data-sharing-mining/> for more details.

Sincerely,

Dr Maurine Neiman

Associate Editor

Board Member

Comments to Author:

The authors have done a great job with the revision, and I thank them for their careful consideration of the comments and insights of both referees, and my own comments, and the resulting modifications to the paper (including addition of random slopes to the models). The paper was sent back to one referee, for expert appraisal, who is very happy with the revised manuscript, but has requested a few minor grammatical fixes prior to finalisation.

Reviewer(s)' Comments to Author:

Referee: 2

Comments to the Author(s).

The revision has been very thorough and addressed my previous concerns, as well as adopted my suggestions on a better logical flow. The paper now reads well and should be of great interest to a wide audience.

However, I noticed a few trivial grammatical mistakes, and occasional tendency in the results to start speculating about causation, which might be better left to the Discussion.

Minor comments:

Ln 182 delete "to" after continued.

Ln 276/277 An example of a sentence drifting into explanation that I would save until the Discussion.

Ln 283 comma needed after "generation".

Ln 292 comma needed after "generations" - unless the rest of the sentence is deleted because it is better moved to the Discussion.

Author's Response to Decision Letter for (RSPB-2021-1843.R0)

See Appendix B.

Decision letter (RSPB-2021-1843.R1)

17-Sep-2021

Dear Dr Travers

I am pleased to inform you that your manuscript entitled "Beneficial cumulative effects of old parental age on offspring fitness" has been accepted for publication in Proceedings B.

Data Accessibility section

Open Access

You are invited to opt for Open Access, making your freely available to all as soon as it is ready for publication under a CCBY licence. Our article processing charge for Open Access is £1700. Corresponding authors from member institutions (<http://royalsocietypublishing.org/site/librarians/allmembers.xhtml>) receive a 25% discount to these charges. For more information please visit <http://royalsocietypublishing.org/open-access>.

Paper charges

Sincerely,

Appendix A

1 10-May-2021

Dear Dr Travers:

I am writing to inform you that your manuscript RSPB-2021-0775 entitled "Beneficial
cumulative effects of old parental age on offspring fitness" has, in its current form, been
rejected for publication in Proceedings B.

This action has been taken on the advice of referees, who have recommended that
substantial revisions are necessary. With this in mind we would be happy to consider a
resubmission, provided the comments of the referees are fully addressed. However please
note that this is not a provisional acceptance.

The resubmission will be treated as a new manuscript. However, we will approach the same
reviewers if they are available and it is deemed appropriate to do so by the Editor. Please
note that resubmissions must be submitted within six months of the date of this email. In
exceptional circumstances, extensions may be possible if agreed with the Editorial Office.
Manuscripts submitted after this date will be automatically rejected.

Please find below the comments made by the referees, not including confidential reports to
the Editor, which I hope you will find useful. If you do choose to resubmit your manuscript,
please upload the following:

1) A 'response to referees' document including details of how you have responded to the
comments, and the adjustments you have made.
2) A clean copy of the manuscript and one with 'tracked changes' indicating your 'response
to referees' comments document.
3) Line numbers in your main document.
4) Data - please see our policies on data sharing to ensure that you are
complying (<https://royalsociety.org/journals/authors/author-guidelines/#data>).

To upload a resubmitted manuscript, log into <http://mc.manuscriptcentral.com/prsb> and enter
your Author Centre, where you will find your manuscript title listed under "Manuscripts with
Decisions." Under "Actions," click on "Create a Resubmission." Please be sure to indicate in
your cover letter that it is a resubmission, and supply the previous reference number.

Sincerely,

Dr Maurine Neiman
mailto: proceedingsb@royalsociety.org

Associate Editor

Board Member: 1

Comments to Author:

In this study, the authors conduct a multigenerational experiment to test the effects of
parental age on offspring fitness in nematodes; presenting results that are at odds with those
predicted by the often-observed Lansing effect (the observation offspring of old parents have
lower fitness in certain taxa). Rather, here the authors find that offspring born to older

parents have higher fitness, and who reach their reproductive peak earlier. The effects
increase in strength over multiple generations of breeding of old age parents, but are
reversible with switching to one generation of breeding by a younger parent.

The paper was reviewed by two referees, each of whom agrees that this study has produced
some fascinating results. But each referee has indicated that major revisions are required.

Referee 1 provided a very thorough review, with many comments that require careful
attention. They noted a major rewrite is required for conceptual precision, and to make the
results clearer (particularly for readers in the *C. elegans* ageing field). This will require a
clearer narrative through the results, which the referee notes as quite terse and disjointed.
The referee also questions the results of the egg size data, since the data is potentially
confounded with the body size data of the young and old parents. Disentangling potentially
confounding effects of egg and parent body size may require the authors to conduct a
supplemental experiment, unless the data are already collected; or otherwise to modify their
inferences.

Referee 2 has noted the Introduction and Discussion are overly long, with too many citations
(for example, the first sentence is followed by 11 citations). They have also provided
suggestions / queries of some of the statistical analyses.

Dear Editor,

Thank you and the reviewers for the valuable feedback provided on our manuscript, and the
opportunity to resubmit. We have now carefully revised the manuscript and made substantial
changes to address the reviewer's comments. Below, we have responded to each comment
and highlight where changes have been made in the MS. We hope it is now suitable for
publication in Proc. B.

On that note, I also have a query on the statistical approach, since I believe it is likely that
the mixed effects models are mis-specified – in terms of the level of replication at which the
parameter estimates are modelled. The denominator degrees of freedom seem to indicate
that the data are modelled at the level of the individual; but I believe the appropriate level to
model the data is the level of the 'ancestral line' – there are 33 ancestral lines; lest the
analyses be pseudoreplicated. Generally, adding a random intercept for "Line ID" will suffice
when it comes to estimation of main fixed effects in a linear mixed model, but the
interactions between fixed effects are still likely to be estimated at the incorrect (individual)
level, leading to inherent pseudoreplication in the models. Correcting for this would require
addition of random slopes (e.g. 1 + propagation regime + generation + propagation
regime:generation|ancestral.line). This same issue would apply for all analyses (including
the survival models, coxme, and including analyses on reversibility of parental effects). For
further information on this issue, please see:

Arnqvist, G. (2020). "Mixed Models Offer No Freedom from Degrees of Freedom." Trends in
Ecology & Evolution.

Thank you for the suggestion to include random slopes. We have tried to incorporate
random slopes for these different models. However, attempting to fit random slopes for the
main effect variables as well as for the interaction is prone to model overfitting, and, in more

practical terms, often results in convergence issues. The main reason for this is that within
unique combinations of ancestral line, treatment and generation, there is no replication, so
the variance for the interaction random slope cannot be estimated.

We fit random intercepts to allow for intrinsic variation of the response variable between
ancestral lines. In addition, we have now modelled random slopes to let the effect of
treatment and generation vary between ancestral lines, except the coxme lifespan model. In
this model, we were unable to fit random slopes for treatment due to convergence issues, so
have only included generation. We have now updated the MS and the full model summaries
to reflect these changes. These additional analyses did not change the results.

Reviewer(s)' Comments to Author:

Referee: 1

Comments to the Author(s)

This is an interesting study of the relationship between aging and fitness in *C. elegans*. It
addresses two questions: how parental age affects offspring fitness, and whether such
effects can accumulate across generations. Parental effects can be positive (maturation
benefits) or negative (senescence effects); transgenerational accumulation of the latter is the
Lansing effect. Or they can be both, as where progeny show increased or earlier
reproduction coupled to more rapid aging. Using careful experimental protocols and well-
designed statistics, the authors detect greater fitness in progeny of older worms, and also a
accumulation of such benefits across generations. The findings are of interest in terms *C.*
*elegans* biology and life history evolution (increases in progeny fitness with parental age
could have an effect on the evolution of aging), and in the context of recent renewed interest
in the Lansing effect. They describe a sort of reverse Lansing effect, presumably resulting
from adult maturational changes (perhaps yolk provisioning of embryos, and changes in
oogenetic mechanisms leading to increased egg size). These results are novel and
intriguing.

While I would support publication of this study in some form, the manuscript needs to be
better presented in terms of clarity of argument, and of presentation of the results, which at
present are difficult to understand in places due to an overly terse style of presentation. In its
present form many readers, particularly from the *C. elegans* lifespan genetics field, will
struggle to understand this manuscript (particularly the results section), limiting its impact
and utility. Another lesser weakness is that the scope of the study is a little limited in terms of
follow up of the findings obtained.

Major points

1) The various phenomena and theories discussed in this study could be presented with a
higher degree of logical clarity and precision. The study deals with several distinct claims
and questions. (1) *bv c* parental age, depending on context (e.g. species, environmental
conditions). A simple interpretation here is that adult maturation can increase progeny fitness
while adult senescence decreases progeny fitness. This underscores the two sides of adult
aging: positive (maturation) and negative (senescence). This could be made clearer. The
presence of these opposite effects is not a contradiction. (2) The Lansing effect refers
specifically to intergenerational deterioration of successive progeny of old parents; Lansing
suggested the hypothesis of an accumulating deteriorative factor, but this hypothesis is
distinct from the Lansing effect. At times in the manuscript, there is insufficiently clear

distinction between (1) and (2). This affect the introduction and discussion, and also the
results section.

Thank you for this comment. We made it clearer that parental age can result in senescence
that decreases offspring fitness and other scenarios that improve offspring fitness. We also
clarified throughout the text the distinction between narrow-sense Lansing effect (the
successive deterioration of the progeny) and the "ageing factor", which is a cause of this
effect as suggested by Lansing.

We note, however, that beneficial effects of parental age do not necessarily result from adult
maturation. For example, they can result from improved access to resources with age and a
relative increase in allocation to reproduction with age, which is why we discuss several
different scenarios and mention key papers in this regard.

We further note that some aspects of Introduction and Discussion go beyond this distinction.
For example, parental age effect is predicted to result in either accelerated or decelerated
evolution of ageing. This refers to the consequences, rather than the cause of parental age
effects.

Finally, there is an interesting interplay between reproduction and survival of the offspring of
old parents, as discussed in a recent review (Monaghan et al. 2020, Trends Ecol Evol). Old
parents can produce short-lived offspring with high early reproduction.

Our study aimed to look at all these questions, which is why they are covered in the
Introduction and discussed further in the Discussion.

2) Line 307-308 in the Results. This is unclear. The issue here, surely, is whether large eggs
give rise to large worms. As I understand it, this is saying that large eggs give rise to worms
that are larger than eggs from young worms. But that could be because progeny of young
worms are smaller, not because the eggs of young worms are smaller. What is needed is a
comparison of egg size vs worm size using eggs from old worms. This needs to be included
in the study, performing additional tests if necessary. It would also be interesting to know
whether other fitness parameters were increased in worms hatched from larger eggs, e.g.
earlier reproductive schedule and increased overall fertility. How does egg size covary with
fitness traits: is it the bigger the egg, the fitter the worm?

Our key interest here is offspring fitness. Larger worms usually produce more eggs, so if old
worms produce larger offspring it fits with our finding that they also produce more eggs.

The second question is how do old worms produce larger offspring? One possibility is that
they produce larger offspring by producing larger eggs that contain more resources. We
measured egg size and, indeed, old worms produce larger eggs. We suggest that offspring
of old worms are larger because they develop from larger eggs and have more resources.
We cannot prove that this is the case, but we suggest that it is quite a reasonable
hypothesis.

We agree that studying the effects of egg size on adult traits is a very interesting question,
but it would require a separate research programme that is beyond the scope of this paper.

Minor points

Line 28. "Such parental effect senescence is" is slightly unclear. "Such parental senescence
effects are"?

Amended as suggested.

Line 29 "higher quality"? We have now changed as suggested.

Line 36. The reasoning in the last two sentences of the abstract is inexact. "contradict the
theory" could imply that this finding argues that the Lansing effect theory is incorrect. But the
Lansing effect is an effect (not a theory) that is seen in various organisms (e.g. rotifers).
More appropriate would be to make the final sentence the penultimate, changing to "that in
C. elegans offspring of" (to avoid the misapprehension that this is a general claim) and the
penultimate the final, remarking that this is the opposite of the Lansing effect (perhaps even
labelling it a "reverse Lansing effect" or something similar.

We have now switched the order of the last two sentences. We suggest that our results
contradict the "ageing factor" theory not the Lansing effect. We also note that it is not clear
whether extinction of 'old parent' lines in rotifers was due to 'ageing' as envisioned by
Lansing, or simply due to a change in fecundity schedule, more in line with our results here
(e.g. King C.E. A re-examination of the Lansing effect. Biology of Rotifers. 1983; 104: 135-
139).

Introduction

Line 44. "many taxa and is the subject of renewed theoretical and empirical interest" Here
"renewed" would help the reader to understand that the Lansing effect is a topic that has lain
relatively dormant for a long time, but has recently reawakened.

Thank you for the suggestion, we have now edited accordingly.

Line 46. "suggested" or "found" ? A point of emphasis: it would be more surprising if
offspring quality did not decline with age, given the presence of parental senescence. What
is surprising about the Lansing effect is the inter-generational effect.

"Found", we have now corrected this.

Line 54. As written it appears that the Lansing effect as a phenomenon is here being
conflated with Lansing's theory about the mechanism underlying the effect. In a similar way,
rate-of-living effects on lifespan do occur, but the rate-of-living theory that rate-of-living
effects are attributable to metabolic rate is incorrect. The authors need to take care to
distinguish the Lansing effect from Lansing's hypothesis about the mechanism involved.

We clarified this by separating the discussion of the phenomenon (the Lansing effect) from
the mechanism (the "ageing factor").

Line 56. The meaning of a decline in developmental time is unclear; would that not be an
increase in fitness? "developmental rate" ?

Yes, thanks for highlighting this error, which is now corrected.

Line 57-58. Has anyone ever tested whether there is a Lansing effect in women? I am
guessing not, and perhaps worth mentioning that it is not know whether such an effect

exists. Risks of some aneuploidies increase with age and can be passed to subsequent
generations, e.g. Downs trisomy 21. Would that count as a Lansing effect?

Yes, for example Gavrilov and Gavrilova 1997 suggested that Lansing effect (in a narrow
sense) is driven by fathers in humans. However, the lines 57-58 in the original MS referred
to the preceding sentence (lines 54-57). There we explicitly state that "in addition to offspring
lifespan, several studies across various taxa show an age-associated 55 decline in several
other traits such as offspring embryo viability, development time, larval and 56 juvenile
survival, and fitness". This is why we discussed reduction in human oocyte quality in lines
57-58.

Line 63-64. "may instead" Again, important to emphasize that what happens depends on
context (species, environment?). "documented. However, in some contexts older parents"
which would mean removing "Indeed" from line 66.

We have now changed as suggested.

Line 78-79. Arguably this will depend on whether maturational or senescent facets of aging
predominate. I think this would be worth arguing here.

The scenarios where parental age positively or negatively influences offspring are discussed
in length in the previous two paragraphs.

Line 79-87. The arguments about evolutionary theory are cogent and apposite.

Thank you.

Line 88-90. "understand", better "explore"? It would be more precise to separate topics (1)
and (2) as described above, rather than conflating them.

We have now changed to "explore".

Line 90. "from young or old parents." ?

Now included.

Line 95. Better "mating with males" ? "cross-fertilization" could imply genotypic differences
between the sexes playing a role here, but the two sexes are generally isogenic.

Now changed to "mating".

Line 106. "cumulative inter-generational effects" ?

Changed as suggested.

Line 114. "wild-type"

Corrected.

Line 115-116. "combine maternal and paternal effects". This is rather inexact.

Hermaphrodites are really self-fertilizing females. Aging (maturation, senescence) will be
affecting oogenesis, and the physiology that supports it, but not spermatogenesis. The male
part is represented only by the presence of the sperm, generated earlier. Thus, paternal
effects are only represented to a lesser extent. A question is whether sperm competition
leaves the worst sperm to last (e.g. nullo X sperm? Male self progeny appear more often
among the last progeny, if I recall correctly).

We agree and have now removed this sentence.

Line 117. "cohorts". "stocks" ? Corrected.

Line 119. "adults in each" Corrected.

Line 121-122. ampicillin, streptomycin, nystatin without capitals. These are not trade names.
Corrected.

Line 126. "0.2 ml E. coli seeding". Rephrase to make this more exact (seeding with an E. coli
suspension?).

We have now amended to be clearer.

Line 156. "haphazardly". "randomly"?

We think "haphazardly" is correct here. Random selection would involve assigning each
worm a unique ID and using a random number generator to select individuals.

Line 158. "egg layings in each generation" ? Corrected.

Line 160. "magnifying." "amplifying"? Corrected.

Line 202. "12x" "120x"? Corrected.

Line 233. "cox proportional hazard", "Cox" (after David Cox)

Corrected.

Results

I found the whole results section too terse and telegraphic. Inclusion of more exposition of
the meaning of the results would improve clarity and readability of this section. Help the
reader to understand what the results mean. It would also help if the subsection headings
more clearly indicated what the data is showing, rather than what is being looked at.

We have now amended the subheadings to be more informative about what we found.

Line 273. The finding that lifespan increased across generations across all parental age
regimes is unexpected, and some sort of remark here is warranted to avoid the bafflement of
the reader, even if only "for reasons that remain unclear" or "for reasons discussed below".
But this results does not seem to be discussed anywhere (or have I missed it?). It needs to
be discussed.

We do not know for sure why there was an increase in lifespan in all parental regimes over
the generations. We suspect a general increase in worm condition due to worms being
housed on individual plates and moved frequently on to new plates, thus reducing negative
effects of overcrowding, food shortages, and infections. Worms may also experience some
negative effects from freezing, which a gradual recovery from could result in improved
condition over successive generations. We have now discussed this in the MS (lines 276-
277, & 290-293).

Line 279. I found this whole section presented in a confusing way. It needs to be made
much clearer what exactly the results are saying and what they mean.

We have now edited this section to further explain what the results mean.

Line 280-282. Here again the difference between (1) and (2) should be spelled out better. I
take it that "proximal effects" refers to (1), but this wants to be clearer.

We have now amended this sentence to be clearer (lines 282-284).

The section overall seems to say, first: that later progeny are fitter, a claim relating to (1).
Then, relating to (2), that intergenerational increases in fitness are seen in both the young
and old groups, arguing against a transgenerational accumulation of increased fitness from
older parents, but this result, like the increase in lifespan, is odd and unexpected: why is
everybody getting fitter? But then it is stated that, after all, fitness is increased
transgenerationally more in the old group. I take it that the point here is that two measures of
fitness are made, and one but not the other indicates transgenerational accumulation of
increased fitness from older parents. The results section needs to include at least indication
of the conclusions that can be drawn from these findings, and how they can be drawn. How
is it that there is not an old parent transgenerational benefit with respect to reproductive
fitness, but for fitness there is?

This is because individual fitness takes into account timing of reproduction.

Line 300-303. Again, the conclusions to be drawn from this section are unclear. The
question is surely, if you compare early progeny of early- and late-propagated worms, is the
greater fitness of the latter still there? Clarity needs to be improved.

We have changed the title of this section to make it clear that accumulated parental effects
can be reversed in one generation.

Line 305. It would be helpful here to describe the change in size, in terms of percentage
change in length and volume, and how the increases in size vary (average change), the size
of the biggest eggs seen. It would be good to include an image showing a normal size egg
and a giant egg.

We have now described the average percentage change in egg size in lines 313-315 (*"On
average, eggs laid by one-day old parents were 7% smaller than those laid by three-day old
worms, while eggs laid on day four were 6% larger than three-day old parents."*).

There are no giant eggs.

Line 305-306. Unclear. I think this means that the larvae emerging from larger eggs develop

more quickly, rather than that the eggs develop more quickly within the hermaphrodite
gonad.

We have changed "eggs" to "larvae" to be clearer.

Line 307. Remove . (period)

Corrected.

Discussion

Line 310. "crucial", hyperbole. "important" ? Changed as suggested.

Line 311. "and has ramifications in", "and is relevant to" ? Changed as suggested.

Line 316. Is that really "with" or should it in fact be "and"? Data should be presented to
clearly establish the "with" (see major point 2 above).

We have now changed "with" to "and".

Line 318-319. "and was maintained throughout six generations" This is ambiguous, and
could give the impression that fitness benefits in progeny of old worms were present for the
subsequence 6 generations regardless of parental age. "and continued to accumulate over
six generations of selection" (unless I have misunderstood something).

We apologise for the confusion; this is now corrected as suggested.

Line 330-331. This is not a cogent conclusion; see my comments above (e.g. on line 36).

We suggest that our results refute the universality of a deleterious 'ageing factor' being
passed on parents to their offspring.

Line 335. "are further supported by", better, "are consistent with"

Changed as suggested.

Line 337. VIT-2 is a component of YP170..

Apologies for the error, this is now corrected.

Line 339-341. The Perez et al study clearly implies this, but it is not made clear here how the
present study does. Surely the increase in eggs size suggest this, and this would be worth
proposing here: the increase in egg size could imply better provisioning, but it doesn't
exactly prove this. Possibly the volume of the embryo is increased to accommodate larger
numbers of yolk and lipid droplets, supporting faster growth within the egg, or perhaps
resulting in emergence of a larger L1 from the egg, with a head start in growth terms.

We have now clarified (lines 345-348).

Line 341. It would be interesting to measure vitellogenin levels in mothers on day 1-4, and in
eggs, in successive generations.

Yes, we agree this would be very interesting but is beyond the scope of this study.

Line 343-344. "the Lansing effect is a result". Surely "Lansing effects can result"

Changed as suggested.

Line 349. "it is conceivable", surely "it is very possible", particularly given the Perez et al
study, among others. Changed as suggested.

Line 353. The argument from here seems highly plausible. It also begs the question: do eggs
from older adults (or larger eggs) give rise to more starvation resistant larvae? Have earlier
studies tested this (at least for eggs from older adults)?

Larger embryos are more starvation resistant (Hibschan et al. 2016).

Line 375. Larger eggs yes, but fewer? Or is this referring to number of eggs laid later by
worms arising from later eggs?

We apologise for the confusion. Egg size increases with age in *C. elegans* while the number
of offspring produced declines (after the reproductive peak on day 2). We have now
removed the sentence "We found that older parents produced larger and fewer eggs than
younger parents."

Line 380-381. Strange statement. The cessation of egg laying is caused by self-sperm
depletion. (Possibly this sentence is intended to express a different idea).

Apologies for the confusion, we have now changed "eggs" to "offspring" to be (lines 386-
387).

Line 382. "all ova". This is not true for lab conditions where many ova remain unfertilized at
the point of sperm depletion; while this remark is justified in the light of later discussion about
wild conditions, qualification here would be appropriate, to avoid confusing the reader.

Fair point, we have removed "all".

Line 383. "suggesting", surely "demonstrating". Changed as suggested.

Line 384-386. Plausible argument. Thank you.

Line 397. "may be part" "appears to be part"? Changed as suggested.

Supplement

Raw data is provided in 6 Numbers files, but a description of the contents of these files
needs to be presented (e.g. in the main Supplementary Materials file).

We have now included a description of the contents in each data file.

One raw data file includes raw data for survival analysis which, if a contents description is
provided, which (as other raw data files) is helpful for anyone wishing to reanalyze the data.
But for the ordinary reader, a table of basic lifespan related statistics should be included in
the usual fashion (showing number of trials, sample sizes, numbers of censored values,
mean lifespan and appropriate statistical comparisons).

We have now included a Lifespan Summary table in the Supplementary materials (Table
S1a).

Referee: 2

Comments to the Author(s)

This is an elegant study. Well designed and adequately replicated. However, I thought the
Introduction could be more focused and the references better distributed to support the
relevant elements/statements (see below). Overall, there are perhaps, too many references.
We have now reduced the number of references in the Intro.

One reason this seems to arise is because of the Discussion spends too long speculating
about the specific limitation of the amount of sperm available in this hermaphrodite.

Ln 43 The first sentence with it's eleven references at the end is not very helpful. It would be
better if the relevant references appeared after "reported in many taxa" and separated from
those relevant to each of "much empirical" and "theoretical interest". On what basis did you
chose the examples in many taxa?

We have now arranged the references as suggested.

We conducted a literature review and selected examples of studies that investigated
parental age effects on offspring.

Ln 52 Insert "further work on pre-industrial " before "humans" Changed as suggested.

Ln 52 start a new paragraph at "The lifespan..." Changed as suggested.

Ln54-57. Unbrigade the references here and distribute them to the relevant trait! Changed
as suggested.

Ln 57 we are back to humans. And is this not a mechanism? So why is the very next
sentence beginning "Several mechanisms, So this new paragraph I'm suggesting goes
Humans, General, Human, General, I suggest you could have a much better, shorter
opening paragraph with a better flow. The subsequent paragraphs Ln 63-77 and Ln 78-87
are much better and then quickly get us to what you did Ln 88.

We have now shortened the introductory paragraph.

Ln 101-105 and 105-107 is repeated later in the Methods. Is it needed here?

We have now shortened these details in the Introduction.

Ln 156 "haphazardly" do you mean "randomly"?

We think "haphazardly" is correct here. Random selection would involve assigning each
worm a unique ID and using a random number generator to select individuals.

Ln 184 why was "very old (four day old) parents" added here for the first time?

Given that Day 4 worms produce so few eggs, it was not practical to include this parental
age in our propagation regime. However, we were curious to know if the increased egg size
with age that we found when we compared Day 1 and 3 worms continued even later in life.
We have now discussed this in the MS (lines 181-182).

Ln 236-240. Are the rates of matricide significantly different between the parental age
regimes?

Yes, there were significantly more matricides in the young parental regime than the old ($\chi^2_1 =$
5.59, $p = 0.02$).

Ln 277 Make it clear both these Chi-sq are for 2 degrees of freedom

We have now included the degrees of freedom.

Ln 288 I think it would be better to present if old parental slope 2.532 is significant rather
than theres no significant difference from young.

The old regime is neither significant from young, or zero ($\beta = 2.532$, $t_{1, 269} = 1.483$ $p =$
0.139). Given that offspring production could increase in worms irrespective of parental
regime (for reasons which we now discuss in the MS lines 290-293) we think that comparing
old to young is more appropriate than comparing old to zero.

Ln 309 - Discussion I would suggest you start with a Summary of your findings. So begin this
section from Ln 313 - "In this study" remove the redundant ", " after "study". The first two
sentences of the current Discussion can be used later.

Thank you for the helpful suggestion, we have now changed as suggested.

Appendix B

10-Sep-2021

Dear Dr Travers

I am pleased to inform you that your manuscript RSPB-2021-1843 entitled "Beneficial cumulative effects of old parental age on offspring fitness" has been accepted for publication in Proceedings B.

The referee(s) have recommended publication, but also suggest some minor revisions to your manuscript. Therefore, I invite you to respond to the referee(s)' comments and revise your manuscript. Because the schedule for publication is very tight, it is a condition of publication that you submit the revised version of your manuscript within 7 days. If you do not think you will be able to meet this date please let us know.

When submitting your revised manuscript, you will be able to respond to the comments made by the referee(s) and upload a file "Response to Referees". You can use this to document any changes you make to the original manuscript. We require a copy of the manuscript with revisions made since the previous version marked as "tracked changes"™ to be included in the "response to referees"™ document.

Online supplementary material will also carry the title and description provided during submission, so please ensure these are accurate and informative. Note that the

Royal Society will not edit or typeset supplementary material and it will be hosted as provided. Please ensure that the supplementary material includes the paper details (authors, title, journal name, article DOI). Your article DOI will be 10.1098/rspb.[paper ID in form xxxx.xxxx e.g. 10.1098/rspb.2016.0049].

In order to ensure effective and robust dissemination and appropriate credit to authors the dataset(s) used should be fully cited. To ensure archived data are available to readers, authors should include a "data accessibility" section immediately after the acknowledgements section. This should list the database and accession number for all data from the article that has been made publicly available, for instance:

• DNA sequences: Genbank accessions F234391-F234402

• Phylogenetic data: TreeBASE accession number S9123

• Final DNA sequence assembly uploaded as online supplemental material

• Climate data and MaxEnt input files: Dryad doi:10.5521/dryad.12311

NB. From April 1 2013, peer reviewed articles based on research funded wholly or partly by RCUK must include, if applicable, a statement on how the underlying research materials "such as data, samples or models" can be accessed. This statement should be included in the data accessibility section.

If you wish to submit your data to Dryad (<http://datadryad.org/>) and have not already done so you can submit your data via this

link [http://datadryad.org/submit?journalID=RSPB&manu=\(Document not available\)](http://datadryad.org/submit?journalID=RSPB&manu=(Document not available)) which will take you to your unique entry in the Dryad repository. If you have already submitted your data to dryad you can make any necessary revisions to your dataset by following the above link.

Please see <https://royalsociety.org/journals/ethics-policies/data-sharing-mining/> for more details.

Sincerely,

Dr Maurine Neiman

Associate Editor
Board Member
Comments to Author:

The authors have done a great job with the revision, and I thank them for their careful consideration of the comments and insights of both referees, and my own comments, and the resulting modifications to the paper (including addition of random slopes to the models). The paper was sent back to one referee, for expert appraisal, who is very happy with the revised manuscript, but has requested a few minor grammatical fixes prior to finalisation.

Dear Editor,

Thank you and the reviewer for your positive response to our revision, and your feedback throughout the reviewing process which has greatly improved the final manuscript. We have now addressed all comments from the current round of review (see below).

Sincerely,
Laura Travers and co-authors

Reviewer(s)' Comments to Author:

Referee: 2

Comments to the Author(s).

The revision has been very thorough and addressed my previous concerns, as well as adopted my suggestions on a better logical flow. The paper now reads well and should be of great interest to a wide audience.

However, I noticed a few trivial grammatical mistakes, and occasional tendency in the results to start speculating about causation, which might be better left to the Discussion.

Minor comments:

Ln 182 delete "to" after continued. Corrected.

Ln 276/277 An example of a sentence drifting into explanation that I would save until the Discussion. We have now removed this sentence from the Results.

Ln 283 comma needed after "generation". Corrected.

Ln 292 comma needed after "generations" - unless the rest of the sentence is deleted because it is better moved to the Discussion. Corrected.